

**Three decades of simulated global terrestrial carbon fluxes from a data**
**assimilation system confronted to different periods of observations.**
Karel Castro-Morales[1*], Gregor Schürmann[1], Christoph Köstler[1], Christian
Rödenbeck[1], Martin Heimann[1,3] and Sönke Zaehle[1,2]
[1] Max Planck Institute for Biogeochemistry, Jena, Germany
[2] Michael-Stifel-Center Jena for Data-Driven and Simulation Science, Jena, Germany
[3] Institute for Atmospheric and Earth System Research, Faculty of Science, University
of Helsinki, Helsinki, Finland
*Now at: Friedrich Schiller University, Institute of Biodiversity, Aquatic
Geomicrobiology, Jena, Germany
**Abstract**
This paper presents global land carbon fluxes for the period 1982-2010 (gross primary
production, GPP, and net ecosystem exchange, NEE) estimated with the Max Planck
Institute – Carbon Cycle Data Assimilation System (MPI-CCDAS v1). The primary
aim of this work is to analyze the performance of the MPI-CCDAS when it is
confronted with three different time periods for data assimilation (DA), and thereby to
assess its prognostic capability. To this extend we assimilated nearly three decades
(1982-2010) of space borne measurements of the fraction of absorbed photosynthetic
active radiation (FAPAR) and atmospheric $CO_2$ concentrations from the global
network of flask and in situ measurements. Both data sets were incorporated with
different assimilation windows covering the periods 1982-1990, 1990-2000 and 1982-
2010. The assimilation results show a considerable improvement in the long-term
trend and seasonality of FAPAR in the Northern Hemisphere, as well as in the long-
term trend and seasonal amplitude of the atmospheric $CO_2$ concentrations when
compared to the observations in sites globally distributed. After the assimilation, the
global net land-atmosphere $CO_2$ exchange (NEE) was $-1.2$ PgC yr$^{-1}$, in agreement
with independent estimates, while gross primary production (GPP; 92.5 PgC yr$^{-1}$) was
somewhat below the magnitude of independent estimates. The NEE in boreal eastern
regions (Northeast Asia) increased on average by $-0.13$ PgC yr$^{-1}$, which translated
into an intensification of the carbon uptake in those regions by nearly 30 % than the
contribution to the global annual average in the model before the assimilation.
Our results demonstrate that using information only over a decade already yielded a
large fraction of the overall model improvement, in particular for the simulation of
phenological seasonality, its interannual variability (IAV) and long-term trend.
Adding longer than decadal data did only lead to very moderate improvements in the
long-term trend of the FAPAR simulated by the model, which may be attributed to the





small model-data mismatch at the long timescales compared to the significantly larger
observational signal and model-data mismatch error at seasonal cycle time scale.
Decadal data also significantly improved the seasonality, IAV and long-term
simulated trend in atmospheric $CO_2$. Importantly, when running the MPI-CCDAS v1
with 30 years of data, the results remained in line with observations throughout this
period, suggesting that the model can represent land uptake to a sufficient degree to
make it compatible with the atmospheric $CO_2$ record. Using data from 1982 to 1990
in the assimilation yielded only a difference to the observations of 2±1.3 ppm for the
period 15 to 19 years after the end of the assimilation. This suggests that despite
imperfections in the representation of IAV, model-data fusion can increase the
prognostic capacity of land carbon cycle models at relevant time-scales.
Key words: *Data assimilation, Global Carbon cycle, modeling, atmospheric $CO_2$.*

**1   Introduction**
The observed contemporary in atmospheric $CO_2$ is driven by anthropogenic emissions
from fossil fuels and land-use change (2007-2016 average: 11.1±0.6 GtC $yr^{-1}$), and
the concurrent net carbon uptake of the ocean and land from the atmosphere, which
take up approximately 22.4 % and 28 % of the anthropogenic flux, respectively (Le
Quéré et al., 2018). Despite recent advances in atmospheric observations, ocean and
land modeling, there remains an imbalance between carbon emissions, ocean and land
sinks, and changes in the atmospheric $CO_2$ concentration of 5.6 % (0.6 GtC $yr^{-1}$).
Despite substantial progress in improving the performance of terrestrial biosphere
models over the past decades, the simulated global terrestrial carbon fluxes and the
net land carbon balance pose still the highest uncertainties from all of the components
of the global carbon cycle (Friedlingstein et al., 2014; Le Quéré et al., 2018).
Quantifying the magnitude and dynamics of the global terrestrial carbon cycle across
different temporal scales and their contribution to the global carbon cycle, is
challenging because the large heterogeneity and complexity of these ecosystems, in
addition to the quantification of contemporary effects and response of these
ecosystems to increasing post-industrial $CO_2$ concentrations (Lienert and Joos, 2018;
Stocker et al., 2014; Wang et al., 2017).
One strategy to reduce the mismatch between the carbon flux predictions of land
surface models and observed trends in atmospheric $CO_2$ concentrations is through
data assimilation (DA) techniques, meaning to "train" the land models by confronting




them systematically with observations of carbon-related variables (Raupach et al.,
2005). During DA, process parameters of land surface models are adjusted through
numerical minimization techniques to reduce the misfit between model results and
actual observations under consideration of the statistical properties of model and
observations. Contrary to the application of atmospheric transport inversion to infer
the sinks and sources of $CO_2$ between the atmosphere and land or ocean from
atmospheric $CO_2$ measurements (Newsam and Enting, 1988; Peylin et al., 2013;
Rayner et al., 1999; Rödenbeck et al., 2003), the application of these carbon cycle
data assimilation systems (CCDAS) provides the additional opportunity to inform the
process-based carbon cycle mechanisms in the land surface model to support a better
estimate and capacity to project dynamics of the terrestrial carbon cycle. Several
CCDAS have been developed for this purpose (e.g. Kaminski et al., 2012; Kaminski
et al., 2013; Lienert and Joos, 2018; Peylin et al., 2016; Scholze et al., 2016).
Although they rely in different statistical methods (i.e. variational or sequential data
assimilation) (Montzka et al., 2012), their common characteristic is integrating long-
term and time consistent global available observational records related to the carbon
cycle such as atmospheric $CO_2$ measurements from flask and in situ networks
(Conway et al., 1994), and remote sensing products of canopy phenology properties
such as MODIS-NDVI (Moderate Resolution Imaging Spectroradiometer -
Normalized Difference Vegetation Index) (Rouse et al., 1974) and FAPAR (Disney et
al., 2016; Pinty et al., 2011a).
In this work, we use the Max Planck Institute - Carbon Cycle Data Assimilation
System (MPI-CCDAS v1, Schürmann et al., 2016) that has been built around the Jena
Scheme Biosphere-Atmosphere Coupling in Hamburg (JSBACH) land-surface model
(Dalmonech and Zaehle, 2013; Raddatz et al., 2007; Reick et al., 2013). The MPI-
CCDAS follows a variational approach that iteratively reduces the model-data misfit
simultaneously for multiple observational and independent carbon cycle data sets
(Kaminski et al., 2013). Since its first development based on the BETHY (Biosphere
Energy-Transfer Hydrology) - CCDAS, the MPI-CCDAS has undergone several code
modifications and improvements, as well as tests of the assimilation of new
observational data sets (e.g. Kaminski et al., 2012; Kaminski et al., 2013; Rayner et
al., 2005; Scholze et al., 2016; Schürmann et al., 2016), with the aim of further



improving the representation of land carbon fluxes. The history of the MPI-CCDAS
and other current DA systems is extensively discussed in Scholze et al. (2017).
In this paper, we seek to analyze the extent to which the application of a CCDAS
leads to the improved representation of the contemporary land carbon cycle and its
prognostic capacity for subsequent years. To this extent, we analyze the estimated
major components of the terrestrial carbon cycle with the MPI-CCDAS in response to
the simultaneous assimilation of three decades of data from two observational
constraints: FAPAR from remote sensing data and atmospheric $CO_2$ concentrations
from the global flask measurements network. Our aim is to analyze the performance
of the MPI-CCDAS to: 1) how well the model is able to reproduce 30 years of
constraint data, 2) how sensitive is the assimilation success to the choice of different
temporal windows used for the assimilation (implying different amounts of
observational data during the assimilation), and 3) how good is the fit to the data in
the time period beyond the period constrained with observations.
**2    Methods**
**2.1    MPI-CCDAS**
The code of the MPI-CCDAS version in this work is identical to the one used in
Schürmann et al. (2016). The model calculates the half-hourly storage and surface
fluxes of energy, water and carbon in terrestrial ecosystems at a coarse spatial
resolution for computational feasibility (8° by 10° grid). The spatial distribution of the
different plant-functional types (PFTs, Table 1) in JSBACH is shown in Fig. S1
(Supplement). The selection of parameters for the assimilation procedure, their prior
values and range was based on Schürmann et al. (2016; Table 1). The initial state of
the parameters was obtained from an independent forward simulation of JSBACH 3.0
(see Sect. 2.3.1). As described in Schürmann et al. (2016), MPI-CCDAS starts with an
initial guess for the model control vector ($\boldsymbol{p}_{pr}$) of e.g. carbon cycle properties, and
model states, and their Gaussian uncertainty ("prior") with covariance $\boldsymbol{C}_{pr}$. The model
control vector $\boldsymbol{p}$ is iteratively updated to minimize a joint cost function $J$ describing
the misfit between observational data-streams ($\boldsymbol{d}$; FAPAR and atmospheric $CO_2$, both
with covariance $\boldsymbol{C}_d$) and the corresponding simulated observation operators of the
MPI-CCDAS $M(\boldsymbol{p})$, taking into account the uncertainties in the observational data
assuming a Gaussian distribution and the information from the prior.
$$J(\boldsymbol{p}) = \tfrac{1}{2}(M(\boldsymbol{p}) - \boldsymbol{d})^T \boldsymbol{C}_d^{-1}(M(\boldsymbol{p}) - \boldsymbol{d}) + (\boldsymbol{p} - \boldsymbol{p}_{pr})^T \boldsymbol{C}_{pr}^{-1}(\boldsymbol{p} - \boldsymbol{p}_{pr}) \qquad (1)$$



During the optimization procedure, a new model trajectory is determined in each
iteration, such that energy and mass are conserved through the entire assimilation
window (Kaminski and Mathieu, 2017). The gradient of the cost function with respect
to the model control vector ($\frac{\partial J}{\partial p}$) is evaluated with a tangent-linear version of JSBACH
3.0, which was generated through automatic differentiation using a TAF
(Transformation of Algorithms in Fortran) compiler tool (Giering and Kaminski,
1998). With this tangent-linear version of the model code, the derivatives for the parts
of the model code where $J(p)$ is evaluated (i.e. code parts that depend on the control
variables), are accurately calculated following the chain rule of calculus. Thus, the
mathematical formulation of the code involved in the cost function must be
differentiable. Since this was not the case for the phenological code of JSBACH 3.0,
the phenology scheme, as described by Schürmann et al. (2016), was updated
following Knorr et al. (2010) where the minimum and maximum calculations in the
entire code were replaced by smoothing functions to avoid steep transitions.
**2.2   Observational data sets**
**2.2.1   FAPAR**
FAPAR is the fraction of the radiation that is absorbed by plants during
photosynthesis, thus is a component of the land-surface radiation budget that
dynamically indicates the status of the vegetation canopy over space and time
(Gobron et al., 2006). In a previous study, MPI-CCDAS was constrained by MODIS-
TIP (Two-stream Inversion Package) FAPAR (hereafter TIP-FAPAR) generated from
the inversion of a 1-D radiation transfer model (Pinty et al., 2006; Pinty et al., 2007)
using the MODIS broadband visible and near-infrared spectral white sky surface
albedo as input (Clerici et al., 2010; Pinty et al., 2011a; Pinty et al., 2011b). For this
study, the TIP-FAPAR product was available only from 2003 to 2011, making it
unsuitable for the indented longer assimilation period. While there are long-term
remotely sensed proxies of FAPAR, such as the NDVI (Rouse et al., 1974), it has
been found previously that NDVI was less reliable that TIP-FAPAR in terms of the
seasonal cycle amplitude of vegetation seasonality (Dalmonech and Zaehle, 2013;
Dalmonech et al., 2015). We therefore merged the Global Inventory Monitoring and
Modeling System (GIMMS) NDVI product, available from 1982 to 2006 (Tucker et
al., 2005), with TIP-FAPAR to provide a longer record of vegetation greenness. The
maximum and minimum NDVI values were rescaled at the pixel level to coincide

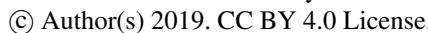



with those from the TIP-FAPAR for the overlapping time periods (2003 to 2006)
following:
$$NDVI_{mod} = \frac{NDVI - NDVI_{min}}{NDVI_{max} - NDVI_{min}} \times (TIP_{max} - TIP_{min}) + TIP_{min} \qquad (2)$$

where TIP is the TIP-FAPAR data. The median uncertainty of the available TIP-
FAPAR data was considered as the uncertainty for the entire time-series. For this
study, this product was aggregated to match the model grid horizontal resolution
considering separately background snow-free and snow-covered conditions
(Schürmann et al., 2016).
As in Schürmann et al. (2016), we applied a mask to the global FAPAR with the aim
of selecting useful pixels in the FAPAR global grid. The selection of pixels to be
removed from the global grid followed three criteria: 1) since no explicit crop
phenology is described in JSBACH, we masked out the grid cells with a crop-
dominated phenology of > 20 %. This step has consequences in areas where other
important functional types are also present in the same grid cells such as deciduous
broadleaves that are also abundant in the USA and Europe. As a result, the parameters
related to deciduous broadleaves are constrained from other locations; 2) we further
masked out pixels that hold a low correlation ($R^2$ < 0.2) when compared the prior
model result and the observations, as we had previously found that the MPI-CCDAS
is incapable of correcting such poor model behaviors (Schürmann et al. 2016); and 3)
finally, we masked out pixels located in areas where phenology abundance is low, i.e.
deserts, because they would influence the optimization causing a large bias due to
compensating effects. The final FAPAR product used during the assimilation contains
only 40 % of the initial number of pixels after the applied mask, resulting in more
pixels distributed in the Northern Hemisphere compared to the Southern areas. This
observational data will be referred hereafter as $FAPAR_{obs}$ (see Fig. 1 for the global
distribution of mean $FAPAR_{obs}$ from 1982 to 2006).
**2.2.2   Atmospheric $CO_2$ concentrations and observation operator**
Measurements of atmospheric $CO_2$ mixing ratios were taken from the flask data
continuous record of 28 sites in the NOAA/CMDL station network (Conway et al.,
1994; Rödenbeck et al., 2003). The selection criteria included length of the record (on
average 19 years) (Fig. A1) and focused on remote and ocean stations with low
impact of local carbon sources and sinks of carbon (Schürmann et al., 2016) (see
location of $CO_2$ stations in Fig. 1). The atmospheric transport of $CO_2$ is calculated in



MPI-CCDAS through the Jacobian representation of the TM3 atmospheric transport
model driven by meteorology fields from NCEP (National Centers for Environmental
Prediction) reanalysis. TM3 is run at horizontal "fine grid" (fg) resolution of $4° \times 5°$
(Heimann and Körner, 2003; Rödenbeck et al., 2003). During the generation of this
matrix representation, the precise sampling time of flask measurements was
considered to minimize the representation error due to short-term fluctuations in
atmospheric transport. The treatment of uncertainties is done in the same way as in
the TM3 atmospheric inversion (Rödenbeck et al., 2003) but, as in Schürmann et al.
(2016), imposing a floor value of 1 ppm to the uncertainties (Rayner et al., 2005) to
allow a range for the comparison to the observational operator.
In order to compare the land fluxes from MPI-CCDAS to atmospheric concentrations,
background carbon fluxes (from fossil fuel emissions, use and change of land cover,
and from the ocean) are necessary to account for the total carbon balance.
*Land-use and land-cover change:* the LULCC fluxes were obtained from a transient
simulation done with the JSBACH 3.0 forced with prescribed annual maps of
modified cover fractions (Hurtt et al., 2006). These fluxes do not consider
disturbances such as fluxes from fires.
*Fossil fuel emissions:* The FF emissions used for this work are the result of a merged
product from various data sets with the aim to complete a long record of emissions,
i.e. 1980 to 2012. This product was prepared for the GEOCARBON project
(www.geocarbon.net) by P. Peylin after merging and harmonizing various data sets:
1) for the period 1980 to 1989, the CDIAC (Carbon Dioxide Information Analysis
Center; http://cdiac.ess-dive.lbl.gov/) product prepared for the CMIP5 exercise
(Andres et al., 2013; Andres et al., 2011; Andres et al., 1996); 2) for the period 1990
to 2009, the IER-EDGAR (Institute of Energy and Rational use of Energy, Stuttgart,
Germany - Emission Database for Global Atmospheric Research;
www.carbones.eu/wcmqs/project/) product where the FF emissions are constructed
using the EDGAR v4.2 data set (http://edgar.jrc.ec.europa.eu/overview.php?v=42)
and completed with profiles for different countries, emission sectors and time zones
available for different temporal resolutions; and 3) for the period 2010 to 2012, the
CarbonTracker product derived at NOAA-Climate Monitoring and Diagnostics
Laboratory (CMDL; https://www.esrl.noaa.gov/gmd/ccgg/carbontracker/).
*Ocean fluxes*: Two products were merged to account for the oceanic $CO_2$ fluxes: 1)
results from the Jena CarboScope v3.4 for the period between 1990-2007 (Rödenbeck



et al., 2013) (http://www.bgc-jena.mpg.de/CarboScope/?ID=s), and 2) ocean annual
fluxes from the Global Carbon Budget 2014 (Le Quéré et al., 2015) (http://cdiac.ess-
dive.lbl.gov/GCP/carbonbudget/2014/). The ocean fluxes for monthly resolution
follow Takahashi et al. (2002) and the spatial distributions follow Mikaloff Fletcher et
al. (2006).
**2.3    Experimental setup**
**2.3.1    Spin up and preparation of initial files**
MPI-CCDAS was forced with meteorology from CRU-NCEP (the Climate Research
Unit from the University of East Anglia, analysis of the NCEP reanalysis atmospheric
forcing) version 6.1. The CRU-NCEP v6.1 reanalysis data is available at daily
resolution from 1901 to 2014 with a spatial resolution of 0.5° (Viovy and Ciais, 2015;
last access July 2015). These atmospheric forcing fields (i.e. wind speed, air
temperature, precipitation, downward short- and long-wave radiation and specific
humidity) were remapped to the coarse (8° × 10°) model grid. In addition, prescribed
annual means (one global mean annual value) of atmospheric $CO_2$ were also included
as    part    of    the    forcing    fields    for    the    model
(https://www.esrl.noaa.gov/gmd/ccgg/trends/global.html, accessed July 2015).
Prior to the assimilation experiments, the JSBACH model was spun up to equilibrium
of the vegetation and soil carbon pools with 1901 atmospheric $CO_2$, land cover and
1901-1910 climate. The spin up procedure was done for a model period of 1000 years
with repeated cycles of atmospheric forcing data. After this period, a transient model
simulation was done also with the forward JSBACH model for the period 1901 to
2012. This transient simulation included change in atmospheric $CO_2$, climate and land
cover. The purpose of this simulation was: i) to obtain the initial conditions for the
CCDAS experiments, and ii) to derive spatially resolved land-use emissions from
JSBACH as additional forcing (see section 2.2.2). Due to technical limitations, the
cover fraction of each PFT is kept constant in MPI-CCDAS during data assimilation,
and thus remained fixed through the simulation period in order to account for the
imprint of the space-time dynamics of land-use change emissions on atmospheric $CO_2$
concentrations. After the spin-up procedure, an initial global scaling factor was set for
the slowly varying carbon pool ($f_{slow}$, also selected as optimization parameter) to
account for non-steady-state conditions at the beginning of the assimilation
(Carvalhais et al., 2008; Schürmann et al., 2016).





### 2.3.2 Assimilation experiments


During the assimilation procedure, the model was forced with the same daily
reanalysis atmospheric data used during the model spin up. The simulation period is
from 1970 to 2010 for all the experiments. In this study we present the results of three
long-term experiments using the MPI-CCDAS, which differ only in the timeframe of
the observational records used during the assimilation: 1) experiment ALL, covers
data in 1980-2010 and includes the complete available timeframe of the two
observational data sets, i.e. for FAPAR is from 1982 to 2006 and for the atmospheric
$CO_2$ concentrations is from 1982 to 2010; 2) DEC1, covers observations available
from 1982 to 1990; and 3) DEC2, covers observations available from 1990 to 2000.
Because of the different lengths of the $CO_2$ records for some stations, this ultimately
leads to different number of observational data used for each experiment (Fig. 2).
In all of the experiments the first ten years of the simulation (1970 to 1979) are to
allow phenology, vegetation productivity and the fast land C pools to adjust to the
new model control vector $p$ and avoid any imprint of the initial conditions on the
calculation of the cost function. The soil C pool at the beginning of the experiment
was included in the model control vector. Thus, the initial period is discarded and
only results from 1980 are reported. The periods of time that fall outside the
assimilation window of the observational constraints on each experiment are thus
periods of model prognosis, i.e. the prognosis period in DEC1 is from 1991 to 2010,
and in DEC2 for 2001 to 2010.

## 3 Results

### 3.1 Mean seasonal phenology

We analyzed the global distribution of FAPAR before and after the assimilation
against the observations. To facilitate the analysis, we also divided the global land
into eight regions: Boreal West and East (BW and BE, for latitudes north of 60 °N),
subtropical Northwest and Northeast (STNW and STNE, between latitudes 20 °N and
60 °N); tropical West and East (TW and TE, between latitudes 20 °N and 20 °S);
subtropical Southwest and Southeast (STSW and STSE, for latitudes south of 20 °S)
(Fig. 1). The normalized RMSE (NRMSE = RMSE / mean(FAPAR$_{obs}$)) and bias
between the modeled and observed FAPAR for 1982 to 2006 are somewhat reduced
by all assimilation experiments compared to the PRIOR (Table 2). One cause for this
decreased model-data misfit is the change in the spatial distribution of LAI, primarily
caused by the optimization of the PFT-specific maximum LAI ($\Lambda_{max}$) parameter (see





Fig. A2 in the appendix for results of parameters changes after the assimilation).
Compared to the PRIOR experiment, the assimilation leads to substantial changes in
the LAI of the tropical forest area, with general reductions of LAI in all three
assimilation experiments. There is less agreement for the extra-tropical areas, with the
ALL experiment suggesting small reductions in LAI, whereas the experiments DEC1
and DEC2 see slight increases (Fig. 3, left panels) relative to the PRIOR experiment.
The second reason for the reduced misfit is an improved representation of the FAPAR
interannual variability (IAV) at a regional scale (Fig. 4) and seasonal cycle at the
pixel level (Fig. S2), particularly in the temperate and boreal zones of the Northern
Hemisphere. This is also evidenced by the increase in linear correlation coefficient $R^2$
between modeled and observed FAPAR with respect to the PRIOR experiment (Fig.
3, right panels). However, it is important to note that the fit remains far from perfect,
likely owing to model structural errors in the way that the meteorological triggers of
phenological events adjust to local climatic conditions. The average global correlation
between model and observations increased moderately in all the assimilation
experiments compared to the PRIOR experiment (Table 2 and Fig. 3). This is
particularly true for the DEC1 and DEC2 experiments ($R^2=0.34$ in both experiments)
than in the experiment that includes all the window of assimilation, i.e. ALL
($R^2=0.20$). The improved correlation is primarily the consequence of an increased
ability of the model to simulate the timing of green-up and brown-down, and its IAV
at regional scale (Fig. 4). Interestingly, the model fit is better if the model is only
subjected to 10 years of data, instead of exposing it to the entire time series.
To further analyze the effect of the assimilation procedure on the simulated
seasonality and monthly growth rate of the FAPAR, we also selected six pixels that
are distributed in locations characterized by a dominant PFT (see Fig. 1 for the
geographic location of the pixels). A clear improvement after the assimilation is in
pixels P1, P2, and P6, where the magnitude of the mean seasonal cycle is better
represented when compared to the observations (Fig. S2). Also, the timing of the
mean seasonal cycle is corrected e.g. in pixels with large seasonal amplitude such as
in P1 (located in Eastern Siberia) and in P6 (located in Canada). While in the PRIOR
experiment (and ALL experiment) the onset and peak of the growing season in P1 and
P6 are delayed by up to two months, in the results from experiments DEC1 and DEC2
this delay is reduced to only one month. This correction might be partially due to
changes in some optimized parameters: increase in the day length at leaf shedding ($t_c$)



and reduction in the temperature at leaf onset $T_\phi$, detected for both the CD and CE
phenotypes (as well as for ETD and TeCr) (Fig. A2); this is because these parameters
control the onset and end of the vegetation activity. This temporal shift however, is
less evident in other pixels such as in P2, despite changes in $T_\phi$ and $t_c$ after the
assimilation in TrH, and this is because the amplitude of the seasonal cycle is small
and only changes in the magnitude of the amplitude are evident (Fig. S2).
In the results of DEC1 and DEC2 for pixel P3 (dominated by TeCr), the water stress
tolerance time ($\tau_w$) and $T_\phi$ were largely reduced, whereas the leaf shedding timescale
($1/\tau_l$; earlier shedding) increased. These changes allowed a considerable improvement
in the timing and duration of the FAPAR in the growing season. The seasonal spatial
distribution of the correlation coefficient $R^2$ for the period 1982-2006 (considering
only the period of available $FAPAR_{obs}$), obtained after the linear correlation between
the $FAPAR_{obs}$ and the model output, is shown in Fig. S3, and the mean global values
are also listed in Table 2. The $R^2$ increased mostly in the Northern Hemisphere where
is evident the spatial extent of the improvement in FAPAR after the assimilation
during spring and autumn in the experiments with a shorter window of assimilation
(DEC1 and DEC2).
**3.2    Mean characteristics of atmospheric $CO_2$**
We next analyze the performance of MPI-CCDAS with respect to the atmospheric
mole fractions of $CO_2$. As example, we compare observed and simulated $CO_2$ mole
fractions at three stations: 1) at the Northern Hemisphere (Alert, ALT), 2) at the
Tropics (Mauna Loa, MLO), and 3) at the Southern Hemisphere (South Pole, SPO) in
terms of the mean seasonal cycle, IAV and monthly growth rate. We also compare the
fluxes from the assimilation to fluxes obtained from an atmospheric transport
inversion (referred to as INV). Very similarly to the MPI-CCDAS, the atmospheric
transport inversion is constrained by atmospheric $CO_2$ data linked surface fluxes
through a tracer transport model, but it adjusts the land surface $CO_2$ fluxes directly
rather than through adjustments to the parameters of a land-surface process model.
The inversion set-up used here is similar to the Jena CarboScope v4.1 (Rödenbeck,
2005; Rödenbeck et al., 2003), involving the same transport model (TM3) as in the
MPI-CCDAS. To make the inversion results as comparable as possible to the results
of this study, the same prior fluxes from fossil fuel emissions and ocean as in MPI-
CCDAS were used, as well as the same $CO_2$ stations. This comparison also helps to





gauge the impact of non-land surface fluxes on the ability to reproduce the
observations. Results on this comparison are shown in Fig. 5. For MLO and ALT the
timing of the seasonal cycle was already well reproduced in the PRIOR simulation,
whereas the assimilation mainly corrects for the amplitude of the seasonal cycle and
the long-term trend. In the SPO there are larger relative differences between the
model results and the observations, however of a much smaller magnitude than for the
two other stations. After the assimilation in the three experiments, the phase in the
seasonal $CO_2$ is shifted by approximately a month to better match the pattern in the
measurements, and the amplitude of the seasonal cycle after the assimilation is in
better agreement with the observations than compared to the PRIOR.
Figure 6 demonstrates that these examples are broadly representative of the global
changes due to the assimilation. Fig. 6a shows that the amplitude for stations located
in the Northern Hemisphere (> 40 °N) is reduced after the assimilation, and in closer
agreement to the observations than in the PRIOR simulation. The largest reduction
took place in the Northernmost Station (ALT) where the seasonal amplitude
decreased from 23.5 ppm in the PRIOR experiment to 16.5 ppm in the ALL
experiment after the assimilation, bringing it closer to the observed amplitude of 14.4
ppm. The latitudinal distribution of the linear correlation coefficient between the
observed and simulated mean seasonal cycles is depicted in Fig. 6b and demonstrates
a very good agreement, i.e. values of $R^2 > 0.9$ in the Northern Hemisphere in all of the
experiments (including the PRIOR simulation). In the tropics (specifically between 40
°N and 20 °N) the MPI-CCDAS achieves an improvement of the model performance
by reducing the misfit of the phasing of the seasonal cycle, as evidenced by an
increased linear correlation (Fig. 6b), however at the expense of reducing the seasonal
cycle amplitude stronger than the observed one. The INV results show a closer
agreement to the observations in the statistical analysis shown in Fig. 5 and 6.
**3.3 Global and regional carbon fluxes**
We next analyzed the spatiotemporal changes of the simulated land surface gross and
net carbon fluxes in the posterior experiments relative to the PRIOR compared to
independent data. At large-scale, the variation of the NBE (net biome exchange of
$CO_2$ with the atmosphere, referred to as the Net Ecosystem Exchange, NEE plus the
land use change related flux) from all of the simulations through the time series is
similar to that from the Global Carbon Project 2017 (GCP17; Le Quéré et al., 2018)
and INV, with the major anomalies collocated in time (Fig. 7a). A comparison of the



fluxes from the ocean and fossil fuels from this data set to the corresponding fluxes
that are prescribed in CCDAS is shown in Fig. 8. The total annual NBE from the three
posterior experiments falls within the spread (shadow green area in Fig. 8d calculated
as ±1 standard deviation) of the NBE mean of the terrestrial ecosystem models in the
GCP17, contrary to the PRIOR simulation. However, the 1982-2010 mean net biome
exchange in all of the assimilation experiments through the time series is on average
1.6 PgC yr$^{-1}$ lower than the flux in the PRIOR simulation ($-2.07$ PgC yr$^{-1}$) and 0.6
PgC yr$^{-1}$ lower than the GCP17 value ($-1.23\pm0.98$ PgC yr$^{-1}$) (Table 3, Fig. 8d and
Fig. S4 for summary of C balance).
In all MPI-CCDAS simulations, the net land-atmosphere C exchange is reduced
relative to the PRIOR result in most of the Southern Hemisphere, while NEE
increased in the Northern Hemisphere (Fig. S5c, e and g). The analysis per regions
illustrates that the extra-tropical northern areas contribute the most to the global net
$CO_2$ flux. The increase in respiration (more $CO_2$ emissions to the atmosphere) in the
tropics is clearly depicted in the latitudinal gradient of NBE shown in Fig. 7c and in
the spatial distribution of the NEE difference between the PRIOR and the posterior
experiments (Fig. S5c, e and g). As in the tropics, the NEE in the southern subtropical
regions was consistently reduced after the assimilation experiments, also switching
the NEE of the STSE region from a C sink of $-0.18$ PgC yr$^{-1}$ in the PRIOR to a mean
C source to the atmosphere of 0.016 PgC yr$^{-1}$ in the DEC2 posterior experiment.
In the boreal east and west regions (BE and BW), the net land C emissions increased
in all of the posterior experiments relative to the PRIOR (Fig. S5c, e and g) with the
largest increase in BE for DEC2 ($-0.29$ PgC yr$^{-1}$) relative to the corresponding value
in the PRIOR ($-0.09$ PgC yr$^{-1}$).
The simulated latitudinal GPP values agree well with the data-driven Model Tree
Ensemble (MTE) estimate from Jung et al. (2011) for the period 1982 to 2010 north
of 30 °N. However, the assimilation results are low biased in the tropics, which
propagated into low estimates of global GPP in all the posterior results (Fig. 7d and
Table 3). After the assimilation, the global GPP and NPP are reduced in the three
posterior experiments compared to the PRIOR (118.5 PgC yr$^{-1}$ and 54.5 PgC yr$^{-1}$,
respectively). In contrast to the posterior global mean of GPP, the value in the PRIOR
simulation compares favorably well to the global mean from the MTE product (118.9
PgC yr$^{-1}$) for the same period of analysis. The global mean GPP is reduced by up to
26 PgC yr$^{-1}$ on average in the three posterior experiments compared to the PRIOR





experiment, but simulation DEC1 experienced the largest reduction in the global
photosynthetic C uptake (mean global GPP of 82.9 PgC yr$^{-1}$) relative to the PRIOR
value (Table 3 and spatial distribution in Fig. S5d, f, and h).
Although the magnitude of the global NBE and GPP is smaller in the posterior
experiments than in the prior, the similar slope detected between the prior and
posterior experiments in the anomaly of these fluxes (calculated relative to the
temporal mean of each time series) (Fig. 7a and b), suggests that the response to the
environmental conditions remains the same through the simulation period even after
the assimilation. This robust response shows e.g. in GPP a similar and gradual
increasing C uptake (positive trend) during the period of analysis, only with a slightly
reduced slope in the PRIOR experiment (Fig. 7b). These trends of the GPP anomaly
differ from the one in the MTE GPP, which is only driven by trends in the remote
sensing FAPAR and climate parameters and it does not consider increases in
photosynthetic light-use efficiency due to $CO_2$ fertilisation.

**3.4    Interannual variability and long-term phenology trend**

We analyzed the long-term trend in the FAPAR signal at regional scale with the
purpose of evaluating if the model is capable of reproducing the observed long-term
greening or browning (i.e. trend to increase or decrease of FAPAR throughout time,
respectively) over large regions (Fig. 1). Compared to the FAPAR$_{obs}$, the IAV of
FAPAR (obtained from the monthly signal for each experiment) is improved only in
the Boreal regions after the assimilation, whereas in the tropical and subtropical
regions, the assimilation does not improve the variability  (Fig. 4). This is also
identified in the monthly growth rate of FAPAR, which is a representation of the
long-term trend for each region after the simulation experiments (Fig. 9). A positive
monthly growth rate indicates a trend for the vegetation to greening, and this is
occurring in all of the regions according to the FAPAR satellite observations, except
in STSW where the long-term trend indicates a decrease of FAPAR (i.e. browning).
In this region, the assimilation improved the long-term trend from a positive to a
negative growth rate in the three posterior experiments (Fig. 9). Despite in most of the
regions the assimilation results agree on a positive long-term trend as in the
observations, the magnitude of this trend is in large disagreement to the observations.
Particularly in the Boreal East region, the PRIOR experiment overestimates the
FAPAR trend by almost double when compared to the observations, and after the
assimilation the trend is reduced leading instead to a slight underestimation of the




growth rate in all of the experiments. The largest disagreement between $FAPAR_{obs}$
and $FAPAR_{mod}$ after the assimilation is in the TW region, where the observations
show a larger and positive trend in FAPAR during the period of analysis, whereas this
trend is not captured in the PRIOR and in all the posterior experiments (Fig. 9).
Despite the lack of correction of variability and long-term trend in most of the
regions, the RMSE between the observations and the regional model results (for
1982-2006) is reduced in all of the areas after the assimilation (Fig. A3). The error
reduction is evident in experiments DEC1 and DEC2 (the average error reduction
after assimilation, in comparison to the error in the PRIOR, for Boreal regions is 19
%, and for subtropical northern and southern regions is about 16 %), except for the
tropical regions TE and TW where the largest error reduction took place after the
ALL experiment (21 % on average) in comparison to the PRIOR experiment.
**3.5   Interannual variability and long-term trends in atmospheric $CO_2$**
During the nearly thirty years of atmospheric $CO_2$ data available, the time series of
the $CO_2$ mole fractions for the PRIOR model underestimate the long-term trend, and
start to deviate in the first five years of the time series; whereas for all the assimilation
experiments, the trend is in closer agreement to the long-term trend of the
measurements during the entire period of the assimilation (leftmost panels of Fig. 5).
This correction to the long-term trend is depicted also in the rest of the stations,
expressed as the linear monthly trend in simulated or observed $CO_2$ concentrations
(rightmost panels of Fig. 5 for ALT, MLO and SPO, and Fig. 6c for summary of all
28 stations). The mean of the growth rate calculated from the results of the ALL
experiment matches very well with the result of the same calculation in the
observations (0.15 ppm month$^{-1}$ in both cases) compared to the PRIOR model (0.087
ppm month$^{-1}$). The results of the experiments using only 10 years of $CO_2$ data show
marked improvements compared to the PRIOR, but tend to underestimate the
atmospheric $CO_2$ trend (thus overestimate the terrestrial land uptake) with DEC1
(0.14 ppm month$^{-1}$) and DEC2 (0.145 ppm month$^{-1}$). Despite moderate improvement,
the MPI-CCDAS is incapable of improving the IAV of the atmospheric $CO_2$
concentration substantially; remaining unchanged the most notable deviations from
the observed signals after the assimilation procedure (Fig. 5).
**3.6   Prognostic capability of MPI-CCDAS**
Finally, we analyze the prognostic capability of CCDAS by comparing the model-
data fit of the decadal assimilation runs to that of the assimilation runs using all data



as a reference case for "best possible" model-data match given the structural
limitations of the MPI-CCDAS to match the observations. To achieve this, we
calculated four-year mean differences between the atmospheric $CO_2$ mole fraction
measurements and the $CO_2$ model results for all of the stations (for the period 1982-
2010) (Fig. 10), and also between the FAPAR satellite data and the monthly FAPAR
model results (for the period 1982-2006) (Fig. 11). We also calculated the RMSE
between the $CO_2$ measurements and model results for each station for four different
periods: 1982-1990, 1990-2000, 2000-2010 and 1982-2010 (Fig. A4). The choice of a
four-year window was made because with Fig. 5 it was established that the capacity
of the MPI-CCDAS to improve the representation of observed interannual variability
was very moderate.
In the ALL assimilation experiment, the atmospheric $CO_2$ concentration is
consistently matched across the entire assimilation period with a $-0.03\pm1$ ppm
average bias to the observations (Fig. 10). This is in striking contrast to the PRIOR
experiment, which fails in all of the stations to reproduce the long-term trend (as
discussed earlier). The four-year mean $CO_2$ mole fraction at the end of this simulation
is 18.8 ppm lower than in the observations. This is also recognized in the RMSE
results where the PRIOR results have the largest error in all of the stations and periods
(between 2.8 and 18.7 ppm) (Fig. A4).
As for the posterior experiments, the performance of the assimilation of $CO_2$ mole
fraction improves, and mostly during the period of the window of assimilation.
During those periods, the difference to the measurements and RMSE is reduced,
whereas the error increases during the periods of time outside of the window of
assimilation. For the DEC1 experiment, the four-year mean difference among the
measurements and the model results is between $-0.3$ and 0.3 ppm in the 80's, a level
at which it remains for the 1990s, where the experiment did not see any observations,
but the fit increasingly degrades after year 2000, with an underestimate of the $CO_2$
mole fraction by 1.6 ppm on a four-year average (still a 90 % reduction in misfit), and
the RMSE is also higher than in DEC2 and ALL for the period 2000-2010 (Fig. A4c).
The model results show that when only the first decade of data is assimilated  (i.e. in
DEC1), a larger deviation to the long-term trend of atmospheric $CO_2$ is identified
after 2000. This is also identified in the results from DEC2 where the lowest four-year
mean difference between the observations and the assimilation results takes place in
the period of the window of assimilation for this experiment (1990-2000) (Fig. 10 and



Fig. A4b for RMSE). During this period, the model overestimates the $CO_2$
atmospheric concentration only by 0.15 ppm on average whereas for the periods of
time outside the window of assimilation, the $CO_2$ concentration is underestimated by
0.64 ppm (in the period 1982-1990) and by 1.04 ppm (in the period 2000-2010).
Thus, also in experiment DEC2 the prognostic skill of CCDAS is reduced outside the
window of assimilation, and the long-term trend is less well reproduced than in the
ALL experiment.
In the ALL assimilation experiment, the atmospheric $CO_2$ concentration is 0±1 ppm
lower than the average value in the observations for the entire simulation period (that
corresponds also to the window of assimilation). This suggests that a longer record of
atmospheric $CO_2$ measurements favorably contributes to a better representation of the
long-term values after the assimilation, but the average deviation to the observations
by using shorter assimilation periods do not deviate far from the upper limit of the
uncertainty when using the longest record.
We also calculated the four-year mean differences between the satellite FAPAR and
the results of the PRIOR and assimilation experiments at regional scale (Fig. 11). In
this case, the prognostic skill of CCDAS for the periods outside the windows of
assimilation is less evident, with a consistent four-year mean difference within the
time series and between experiments.
**4    Discussion**
The simultaneous assimilation of long-term space borne FAPAR and atmospheric
$CO_2$ measurements in the MPI-CCDAS leads to an overall improvement in the
modeled global carbon fluxes (as summarized in Fig. A3 and A4). The MPI-CCDAS
is capable of extracting information about the seasonal cycle and the long-term trend
from the FAPAR observations. However, the imprint of the interannual variability
(IAV) on the cost function of the MPI-CCDAS is comparatively low. Therefore, the
IAV remains largely unchanged in the posterior. With the exception of the tropical
latitudes, the mismatch between observations and model output is small, and thus of
little concern. The lacking ability of the MPI-CCDAS to reproduce the higher IAV in
the tropical bands, may be indicative of a too weak drought response in the maximum
leaf area index of the model. However, the modeled signal remains within 0.05
FAPAR (dimensionless) of the observations, and the importance of this mismatch
should thus not be too interpreted too strongly.



The use of decade-long FAPAR data (DEC1 and DEC2) already leads to notable
improvement of the simulated seasonal phenology of the land surface. This
improvement is predominantly the result of adjustments in the Northern Hemisphere
dominated by phenotypes controlled by parameters for temperature and day-length
thresholds such as deciduous and evergreen needle leaf and extra-tropical deciduous
trees. Thus the optimization of parameters that regulate the onset and end of the
growing season improved the timing of the Northern Hemisphere FAPAR during
spring and autumn. This finding is generally consistent with the previous application
of the MPI-CCDAS for only 5 years (Schürmann et al., 2016).
A long-term greening trend in vegetation, especially in boreal regions has been
previously observed in analysis of space borne data (Forkel et al., 2016; Lucht et al.,
2002). While this enhanced vegetation greening was captured in the model already
before the assimilation, it was mostly overestimated in northern regions and
underestimated in the Southern Hemisphere. At regional scale, the assimilation in all
of the posterior experiments improved the growth rate of FAPAR, reflecting a
greening trend, and is in closer agreement to the satellite FAPAR data. This was
mostly achieved in boreal regions. However, the moderate improvements in the
simulated trend in temperate regions of the western hemisphere are associated with a
decreased performance in the eastern hemisphere, indicating that the model structure
of MPI-CCDAS is incapable of reconciling regional differences. It is unclear whether
this is an indication of the need to parameterize these hemispheres differently in terms
of their phenological response to the underlying driving factors (such as temperature,
moisture availability and day-length), or whether land-use or vegetation dynamics
processes not considered by MPI-CCDAS are the reason for this mismatch. Despite
these broad-scale improvements, at pixel level the MPI-CCDAS does not necessarily
reproduce the magnitude of the greening trend and its interannual variability in all the
posterior experiments, which results from the structural dependence of the MPI-
CCDAS on few, globally applicable PFT-level parameters, and challenges in using
the spatial mixed signal at the model resolution to infer PFT-specific parameters. A
likely better strategy for constraining these PFT-specific parameters would be to
resample the highly resolved satellite product to PFT-specific FAPAR maps prior to
aggregation, and provide PFT-specific FAPAR maps to the CCDAS.
Our results also demonstrate that the long-term trend of atmospheric $CO_2$ and of its
seasonal amplitude in the Northern Hemisphere and at station level is considerably





improved. This is independent of the different periods of data used for the
assimilation. However, the MPI-CCDAS consistently fails to resolve some of the
features of the year-to-year variability of the measured atmospheric $CO_2$ stations,
which translates into an acceptable, but far from perfect fit to the inferred annual
carbon budget of the global carbon project (Le Quéré et al., 2018). We compared the
performance at this time-scale to the results from an atmospheric $CO_2$ inversion (INV)
with the same input fields and atmospheric transport model than MPI-CCDAS, to
illustrate that these deviations do not reflect uncertainties in the representation of the
atmospheric transport. It needs to be borne in mind that both the choice of the
atmospheric transport model (and associated imperfections at resolving the vertical
and lateral atmospheric transport of $CO_2$) and the method to aggregate atmospheric
observations to obtain an estimate of the annual growth rate in the global carbon
budget introduce some error in any estimate of the interannual variability. As a
consequence, only the occurrence of larger model-data mismatches is of concern and
a can be interpreted as a genuine result of the MPI-CCDAS' inability to correctly
resolve the carbon flux variation.
Particularly, the model is not able to capture large-scale relevant climatic disturbances
that influence the interannual variability of the carbon cycle like fires, or the decrease
in atmospheric $CO_2$ growth after explosive volcanic eruptions such as for Mt.
Pinatubo in 1991, or increase in atmospheric $CO_2$ concentration due to fire occurrence
associated with El Niño events (Frölicher et al., 2011; Frölicher et al., 2013). MPI-
CCDAS lacks a representation of the effect of diffusive radiation on photosynthesis
that likely contributed to the post-Pinatubo increase in terrestrial carbon uptake
(Mercado et al., 2009). Other important limitations in the current MPI-CCDAS
structure that influence the results are: the possibility of only prescribe annual non-
dynamic LUCC fields, limiting the performance of the model for long-term dynamic
changes in vegetation (Reick et al., 2013) and the possibility to dynamically account
for fire disturbance (Lasslop et al., 2014) and peatland fires.
Independently of the amount of data used in the assimilation window, our results
show that the GPP and NEE were consistently reduced globally compared to the prior
run, i.e. less carbon uptake by plants leading to the model results to be in closer
agreement to other independent estimates such as the GCP17. The MPI-CCDAS
suggests a somewhat lower average annual atmospheric $CO_2$ growth rate (calculated
by the sum of the net C fluxes from the ocean, land and fossil fuel emissions) than the





one estimated in the GCP17 (Le Quéré et al., 2018), even if the MPI-CCDAS estimate
falls within the uncertainty of the GCP17 (Fig. 8 and S4). Most of the difference
stems from small differences in the assumed fossil and ocean carbon fluxes. In case of
the carbon fluxes from fossil fuels, the data prescribed in MPI-CCDAS does not
contain fluxes due to e.g. cement and flaring, thus the magnitude of the annual carbon
sources through the time series is consistently somewhat lower but still within the ±5
% uncertainty of the GCP17 data (Le Quéré et al., 2018). As for the ocean carbon
sink, the annual mean values prescribed in MPI-CCDAS are also of lower magnitude
than the mean value in the GCP17 but falling in the lower limit of the uncertainty
value (Fig. 8c and S4). The flux due to LULCC prescribed in MPI-CCDAS is also of
lower magnitude than that one from the GCP17 because the simulation made by
JSBACH 3.0 does not consider disturbances like fires and gross transitions, which
might have also contributed to the lower land C sink obtained in the assimilation
experiments compared to the total land C sink in GCP17.
Compared to independent estimates of GPP (Jung et al., 2007), the MPI-CCDAS GPP
matches well in regions with a distinct, light and temperature driven seasonal cycle
(i.e. north of approx. 30 °N), translating to a reduction in modeled GPP by 0.7 PgC
$yr^{-1}$ in boreal regions. However, as in Schürmann et al. (2016), the tropical
productivity is strongly reduced by the assimilation to estimates that are substantially
lower than independent estimates (Jung et al., 2007). An important factor influencing
the global reduction of GPP and the tropical uptake of C appears to be related to the
difference in data availability of $CO_2$ stations between the assimilation windows,
specifically the fact that in the data-poor period DEC1, topical GPP is substantially
lower than estimated independently and compared to the assimilation runs with more
stations in DEC2 and ALL. As a result, the mean tropical land C source to the
atmosphere in the prior experiment (mean NBE value of 0.12 PgC $yr^{-1}$, and minimum
value of −0.07 PgC $yr^{-1}$, reflecting C uptake in the 4 °S latitudinal band) was
increased to 0.37±0.17 PgC $yr^{-1}$ on average for all the posterior results.
The total global vegetation C stock in all of the experiments, including the PRIOR, is
in closer agreement to the lower end of the estimate by Carvalhais et al., (2014) (296
PgC). In the posterior experiments, the vegetation C pool decreased between 14 and
20 % of the value in the PRIOR but still remaining within the range of the literature
estimate (442±146 PgC). The global soil C stock showed a more drastic change after
the assimilation. In all the posterior experiments, the soil C pool decreased by 45, 43



and 53 % with respect to the value in the PRIOR. Particularly after the assimilation,
the total C in the soil (1362 PgC) in the ALL experiment is in closer agreement to the
estimate      from      the      Harmonized      World      Soil      Database
(http://webarchive.iiasa.ac.at/Research/LUC/External-World-soil-database/HTML;
last access January 2015) of 1343 PgC (Table 3). It is important noting that the
JSBACH 3.0 version used in this MPI-CCDAS does not include permafrost
processes; therefore the global soil C stock might be underestimated.
The parameter optimisation resulted in considerable reduction in the cost function and
norm of gradient, which can be clearly seen as a reduction in the root mean squared
error of the MPI-CCDAS compared to the FAPAR and $CO_2$ observations (Fig. A3
and A4). The trajectory of model parameters involved in the optimization differed for
each experiment and each phenotype. While some parameters such as the maximum
leaf area of grasses and shrubs and the correction parameter for the initial soil pool
size were consistently retrieved, some final parameter estimates varied considerably
between the three experiments, e.g., the tropical maximum leaf area index and some
of the parameters controlling the seasonality of the phenology (Figure A2). The
consequence of these variations are regional differences in the simulated compartment
fluxes GPP and ecosystem respiration, which apparently are not well constrained
from the observations. Interestingly, these differences lead to very similar absolute
values in global carbon fluxes and their trends. This clearly demonstrates a certain
degree of equifinality in the results, and cautions a too stringent interpretation of the
outcome of the MPI-CCDAS in terms of improving understanding about biosphere
processes and their long-term trends.
Notwithstanding these conceptual issues, the set-up of this study enables to test by
how much the quality of the data-model agreement is reduced by exposing the MPI-
CCDAS to shorter observational time-series. This can be done by comparing the
results of the ALL experiment to the years of 1990-2010 for the DEC1 experiment,
and for 2000-2010 for the DEC2 experiment. In terms of FAPAR, there is no clear
degradation of fit with time even though in general terms the trend in the data are best
matched with the ALL experiment. This is foremost a consequence of comparatively
small trends in observed FAPAR, implying that extracting mean seasonal patterns and
amplitude for a few years is most essential for simulating current and near-term
FAPAR. This would suggest that a focus of assimilation on high-quality and highly
spatially resolved FAPAR should be a priority over the use of long-term data sets.





The results are different for the case of projecting atmospheric $CO_2$, where the model-
data agreement of approximately ±0.5 ppm during the assimilation period starts to
deviate for the DEC1 experiment later than 10 years after the end of the assimilation
window, whereas in the DEC2 experiment, the degradation of the model-data match
already starts after approximately 5 years. Nonetheless, with the caveat that MPI-
CCDAS does not fully explain the interannual variability of the land net carbon flux,
this suggests a reasonable short-term forecasting (for a small number of years) skill of
atmospheric $CO_2$.
**5    Conclusion**
The MPI-CCDAS is capable of simultaneously integrating two independent
observational data sets over three consecutive decades at the global scale to estimate
global carbon fluxes. The results demonstrate that assimilating only one decade of
observations, for two observational data (FAPAR and atmospheric $CO_2$
concentrations), leads to broadly comparable results and trends in the global carbon
cycle components than using the full time series of available observations (thirty
years). Currently the system is able to confidently predict the carbon fluxes in short
time scales (up to 5 years after the end of the window of assimilation) e.g. for
atmospheric $CO_2$ concentrations at site level, and the mean prediction remains within
the uncertainty of the observations. However, long-term predictions with CCDAS are
more uncertain, as the observational record does not fully constrain the long-term land
net C uptake in the current phase of rising atmospheric $CO_2$ and gradually changing
climate. The MPI-CCDAS is a computational expensive system, and the
demonstration that large-scale carbon fluxes can be improved by only using a limited
period of observations increases the feasibility of using DA to constrain the land
carbon budget in land surface models. However, we also show that there is
considerable variations in the estimated parameter space and regional distribution of
the land C uptake suggesting that further improvements in the land-surface model,
especially in the current structure and design, must be first solved to improve the
model and computational efficiencies of the system before an attempt to include
another observational stream can be made to potentially improve its prognostic skill.



**6 Code availability**

The code of the JSBACH model is available upon request to S. Zaehle (szaehle@bgc-jena.mpg.de). The TM3 model code is available upon request to C. Rödenbeck (christian.roedenbeck@bgc-jena.mpg.de). The TAF-generated derivative code is not available and it is subject to license restrictions.

**Acknowledgements**

This research was supported by the European Space Agency through the STSE Carbonflux (contract no. 4000107086/12/NL/Fv0), the 7[th] Framework program of the European Commission (grant no. GEOCARBON FP7-283080), as well as the Max Planck Society for the Advancement of Science, e.V., through the ENIGMA project. The authors thank P. Peylin for providing the fossil fuel emission data and T. Thum for the constructive comments during the preparation of the manuscript.





Table 1 – Model parameters selected for the optimization: rows 1 to 6: related to phenology, row 7 to photosynthesis and rows 8 to 11 to land-carbon turnover. The values in the table for each PFT (where applies only) are for the prior conditions: $p_{pr}±C_{pr}$. &Values in $f_{photos}$ are the photosynthetic parameters $Vc_{max} / J_{max}$ (μmol CO₂ m⁻² s⁻¹ / μmol m⁻² s⁻¹). In $Λ_{max}$ the values marked with * are multiplied in the model by a factor of 1±0.2 and those with ^ (in $Λ_{max}$ and in $f_{photos}$) by a factor of 1±0.1; in $f_{photos}$ values with [a] are multiplied by 1±0.02, [b] by 1±0.03 and [c] by 1±0.06; these operations allowed a change in the standard values in the model. Letters in parenthesis below each PFT name are the predominant environmental controls that influence each group: T, temperature; D, daylight; W, water.

| Parameter | Description | TrBe (W) | TrBD (W) | ETD (T,D) | CE (T,D) | CD (T,D) | RS (W) | TeH (T,W) | TeCr (T,W) | TrH (T,W) | TrCr (T,W) |
|---|---|---|---|---|---|---|---|---|---|---|---|
| $Λ_{max}$ | Maximum LAI (m² m⁻²) | 7.0* | 7.0* | 5.0* | 1.7* | 5.0* | 2.0* | 3.0^ | 4.0^ | 3.0^ | 4.0^ |
| $1/τ_l$ | Leaf shedding timescale (d⁻¹) | | | 0.07±0.01 | 5e-4±1e-4 | 0.07±0.01 | 0.07±0.01 | 0.07±0.01 | 0.07±0.01 | 0.07±0.01 | 0.07±0.01 |
| $τ_w$ | Water stress tolerance time (d) | 300±30 | 114±10 | - | - | - | 50±5 | 250±25 | 250±25 | 250±25 | 250±25 |
| $T_φ$ | Temperature at leaf onset (°C) | - | - | 9.21±1 | 9.21±1 | 9.21±1 | - | 1.92±0.5 | 1.92±0.5 | 1.92±0.5 | 1.92±0.5 |
| $t_c$ | Day length at leaf shedding (h) | - | - | 13.37±1 | 13.37±1 | 13.37±1 | - | - | - | - | - |
| $ξ$ | Initial leaf growth state (d⁻¹) | | | | | 0.37±0.03 | | | | | |
| $f_{photos}$ & | Photosynthesis rate modifier | 39.0/ 74.1^ | 31.0/ 58.9^ | 66.0/ 125.4[a] | 62.5/ 118.8[b] | 39.1/ 74.3[c] | 61.7/ 117.2^ | 78.2/ 148.6^ | 100.7/ 91.3^ | 8.0/ 140.0^ | 39.0/ 700.0^ |
| $Q_{10}$ | Temperature sensitivity to resp. | | | | | 1.8±0.15 | | | | | |
| $f_{slow}$ | Multiplier for initial slow pool | | | | | 1±0.1 | | | | | |
| $f_{aut\_leaf}$ | Leaf fraction of maintenance resp. | | | | | 0.4±0.1 | | | | | |
| $CO_2^{offset}$ | Initial atmospheric carbon (ppm) | | | | | 0±3 | | | | | |

TrBE, Tropical evergreen trees; TrBD, Tropical deciduous trees; RS, Rain-green shrubs; CE, Coniferous evergreen trees; ETD, Extra-tropical deciduous trees; CD, Coniferous deciduous trees; TeH, C3 grasses; TeCr, C3 crops; TrH, C4 grasses; TrCr, C4 crops.





Table 2 – Statistical analysis of FAPAR for 1982 – 2006 in all of the experiments, and also for the periods of the window of assimilation only for DEC1 and DEC2. $R^2$ is obtained from the linear correlation between $FAPAR_{obs}$ and $FAPAR_{mod}$ calculated for the entire period and by seasons.

| | Bias | NRMSE | $R^2$ | | | | |
|---|---|---|---|---|---|---|---|
| | | | All year | DJF | MAM | JJA | SON |
| PRIOR | 0.37 | 0.95 | 0.16 | 0.14 | 0.31 | 0.21 | 0.33 |
| ALL | 0.10 | 0.76 | 0.20 | 0.14 | 0.34 | 0.20 | 0.37 |
| DEC1 | 0.08 | 0.64 | 0.34 | 0.15 | 0.39 | 0.18 | 0.41 |
| DEC2 | 0.09 | 0.65 | 0.34 | 0.14 | 0.39 | 0.18 | 0.41 |
| Only for the period of the assimilation window | | | | | | | |
| DEC1 (1980-1990) | 0.09 | 0.66 | 0.34 | 0.18 | 0.42 | 0.21 | 0.48 |
| DEC2 (1990-2000) | 0.05 | 0.48 | 0.34 | 0.18 | 0.41 | 0.21 | 0.47 |

Table 3 – Global average of the terrestrial carbon cycle components and carbon stocks in results from the assimilation experiments and prior (1980-2010), and other independent estimates (see table foot for description).

| | PRIOR | ALL | DEC1 | DEC2 | INV | Literature |
|---|---|---|---|---|---|---|
| GPP (PgC yr$^{-1}$) | 118.5 | 96.8 | 82.9 | 97.0 | - | 118.9[a] |
| NPP (PgC yr$^{-1}$) | 54.5 | 34.2 | 37.2 | 30.2 | - | - |
| NEE (PgC yr$^{-1}$) | −2.65 | −1.14 | −1.32 | −1.17 | −1.17[c] | −2.25±1.17[b] |
| NBE (NEE + LUCC) (PgC yr$^{-1}$) | −2.07 | −0.56 | −0.74 | −0.59 | - | −1.23±0.98[b] |
| ER (PgC yr$^{-1}$) | 115.5 | 95.0 | 80.9 | 95.1 | - | - |
| Ra (PgC yr$^{-1}$) | 64.1 | 62.6 | 45.7 | 66.8 | - | - |
| Rh (PgC yr$^{-1}$) | 51.4 | 32.4 | 65.2 | 28.3 | - | - |
| Soil C (PgC) | 2480 | 1362 | 1422 | 1165 | - | 1343[d] |
| Vegetation C (PgC) | 392 | 311 | 335 | 312 | - | 442±146[e] |
| Litter C (PgC) | 228 | 167 | 171 | 158 | - | - |

[a] Model Tree Ensemble data-driven product; Jung et al., 2011; average for 1982-2010,
[b] Global Carbon Project 2017; Le Quéré et al., 2018; average for 1980-2010. The NBE values include the LULCC reported for each individual model.
[c] Inversion result is the average for 1980-2009
[d] http://webarchive.iiasa.ac.at/Research/LUC/External-World-soil-database/HTML
[e] Carvalhais et al. (2014).



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





Figures.

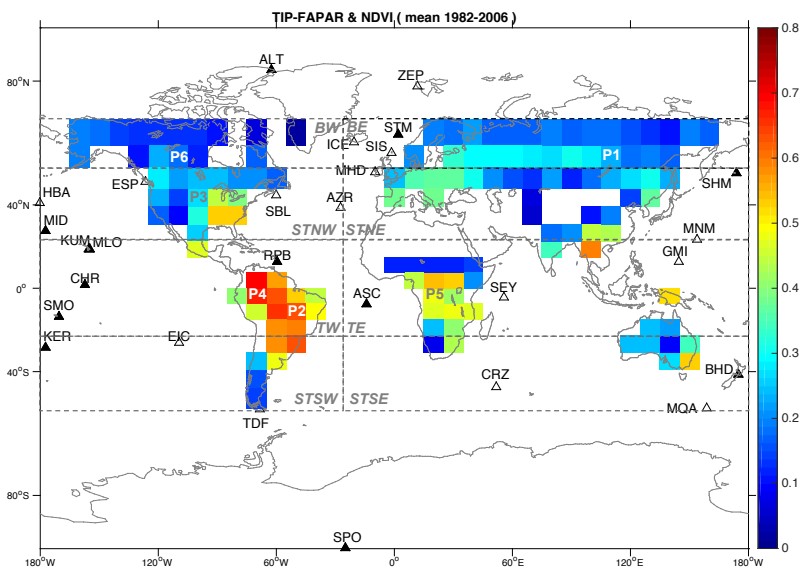

Figure 1 – Global distribution of the temporal mean (1982-2006) of the merged satellite FAPAR product used in the assimilation procedure. It shows also the spatial coverage of eight regions globally distributed: Boreal West and East (BW and BE, for latitudes north of 60 °N), subtropical Northwest and Northeast (STNW and STNE, between latitudes 20 °N and 60 °N); tropical West and East (TW and TE, between latitudes 20 °N and 20 °S); subtropical Southwest and Southeast (STSW and STSE, for latitudes south of 20 S). Also shown six selected pixels: P1, for the coniferous deciduous (CD) phenotype in the East Siberian Taiga; P2, for the C4 pastures and grasses (TrH) of central Brazil; P3, for the C3 and C4 crops, pastures and grasses (TeCr and TeH) of Northern USA; P4 and P5, for tropical evergreen trees (TrBe) situated in Northwestern Brazil and central Africa; and P6, for coniferous evergreen (CE) located in Canada; and the location of 28 stations of the $CO_2$ network measurements (filled triangles, stations only included in DEC1; empty triangles, stations included also in ALL and DEC2) for analysis of the assimilation results.



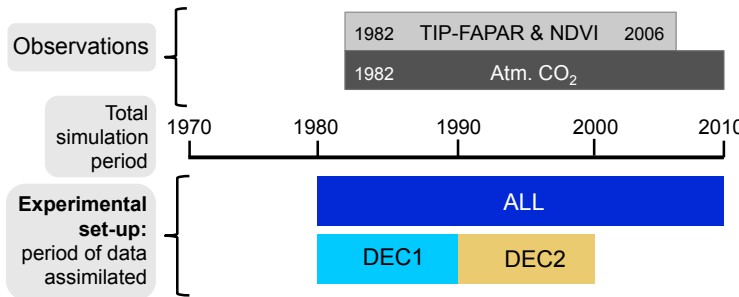

Figure 2 – Experimental set up for posterior experiments ALL, DEC1 and DEC2 that use different temporal windows for the assimilation of observations of FAPAR and molar fractions of atmospheric $CO_2$.

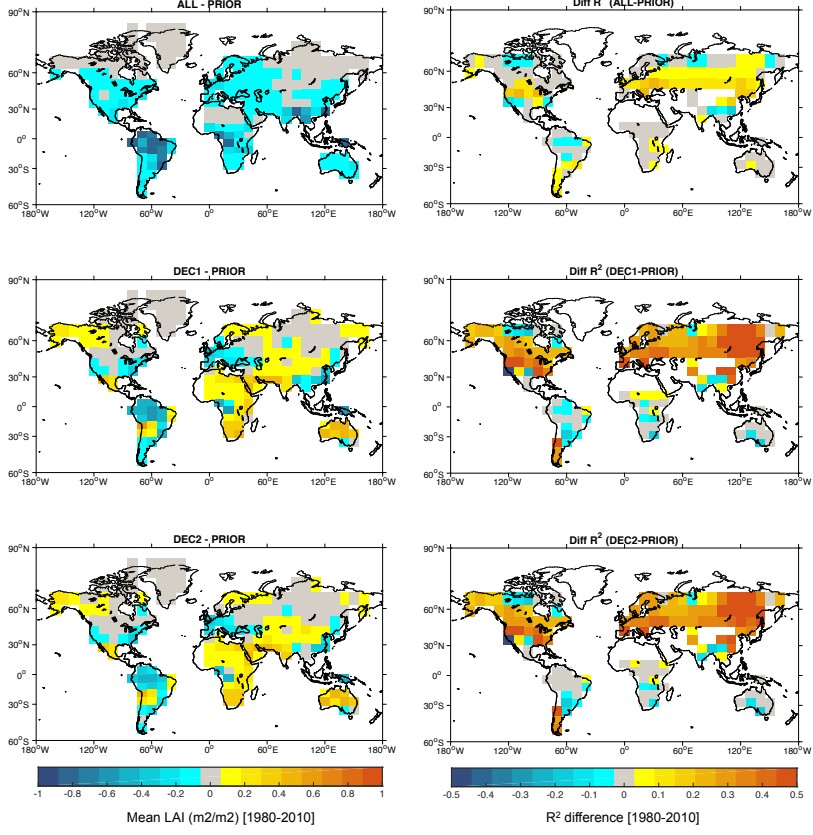

Figure 3 – Spatial difference between the results from the posterior and the PRIOR experiments for the total period of the simulation (1980-2010) of the mean Leaf Area Index (LAI) (left panels) and the correlation coefficient ($R^2$) of FAPAR between the model and the observations (right panels).





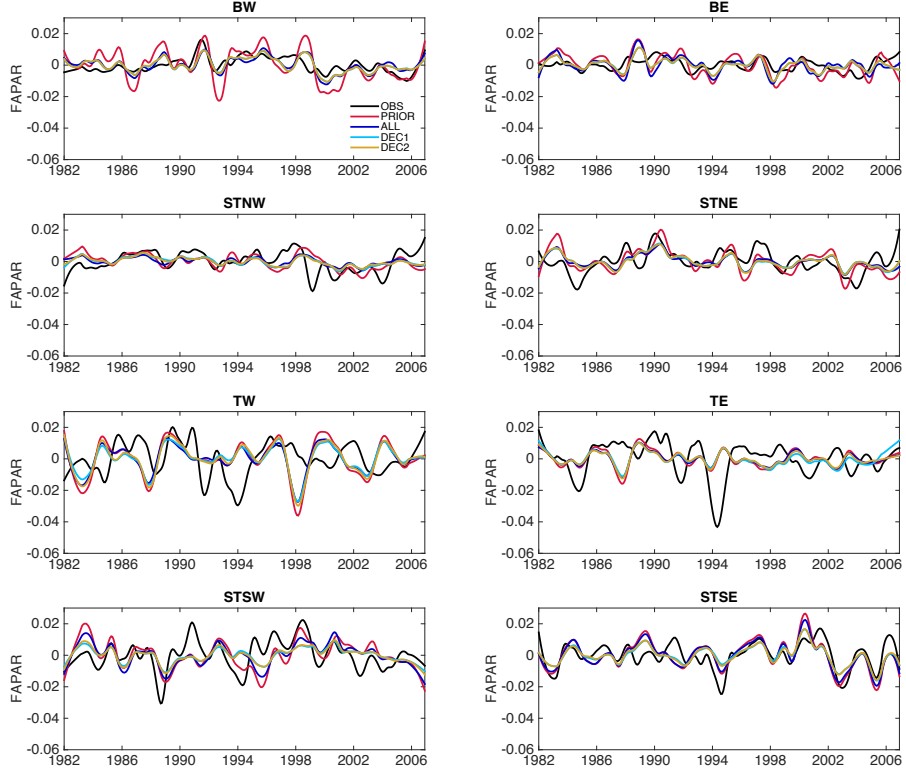

Figure 4 – Interannual variability of FAPAR in the satellite observations and model experiments for the six selected regions.



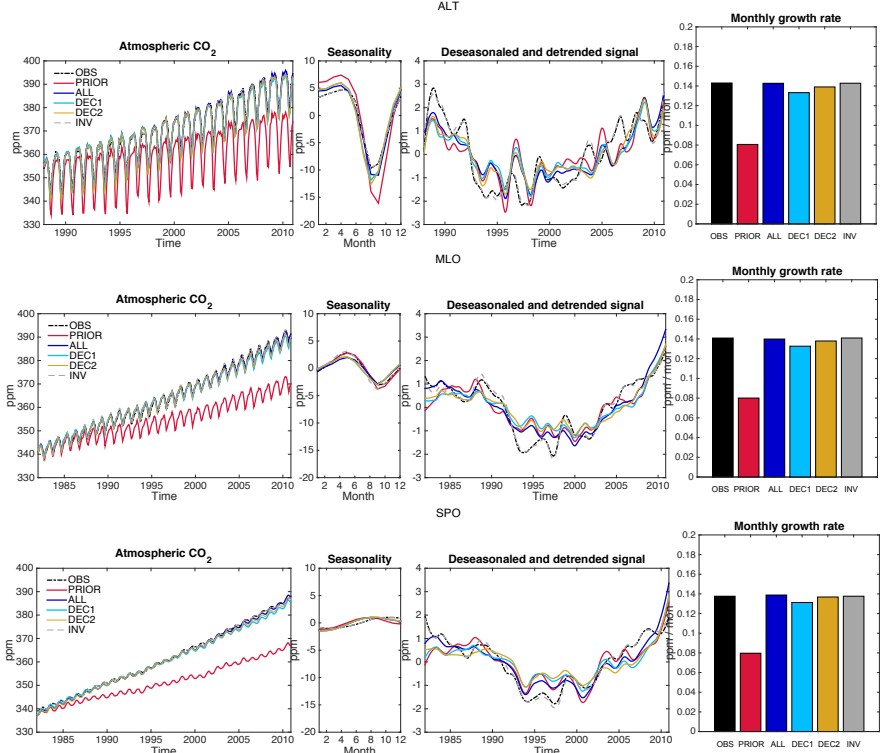

Figure 5 – Statistical analysis of atmospheric $CO_2$ in three flask measurement sites: Alert (ALT; top panels), Mauna Loa (MLO, center panels) and South Pole (SPO, bottom panels), from the measurements, PRIOR, posterior experiments (ALL, DEC1 and DEC2) and inversion (INV1). For each station the panels show the time series of the mean monthly values, the mean seasonal cycle, the interannual variability and the monthly growth rate for the entire period of the simulation (1980-2010).



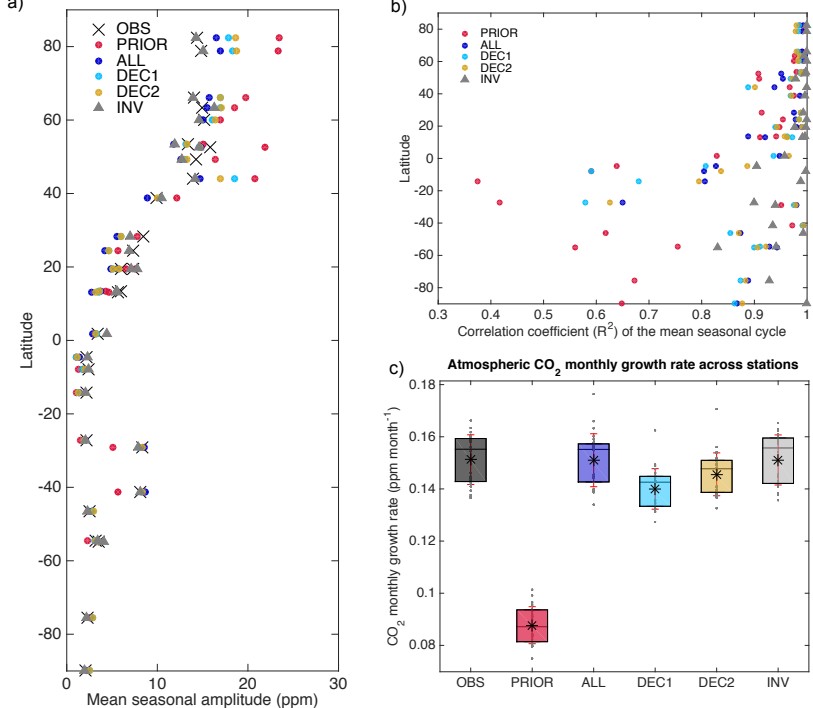

Figure 6 – a) Latitudinal distribution of the mean $CO_2$ seasonal amplitude for the 28 flask-measurement stations from the observations, PRIOR and posterior experiments; b) Latitudinal distribution of $R^2$ obtained from the correlation between the observations and each simulation results of the mean atm. $CO_2$ seasonal cycle and c) average atmospheric $CO_2$ monthly growth rate across stations for the observations and model results. The star on each bar is the mean of the atm. $CO_2$ monthly growth rate, the horizontal middle black line on each box is the median, the red whiskers depict the error as $+/- 1\sigma$, and the grey dots on each box are the actual monthly growth rate values for all the stations in each data set.





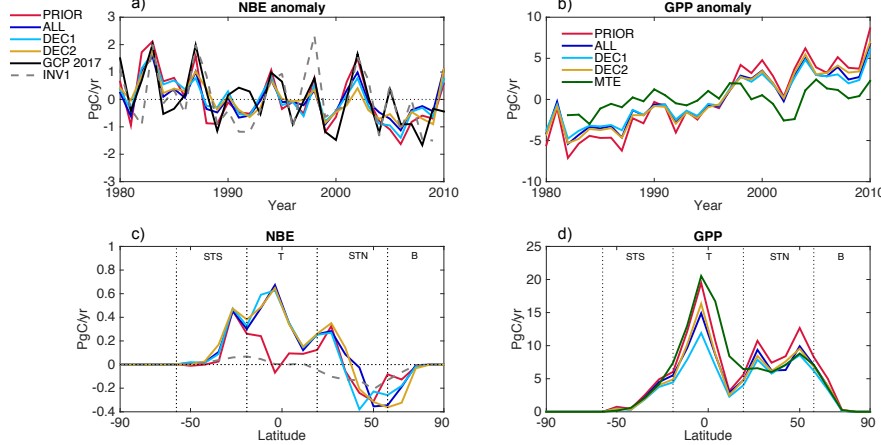

Figure 7 – Time series of the anomaly to the temporal mean of the time series (a and b), and latitudinal gradient (c and d) of the total Net Ecosystem Exchange (NEE including the influence of LULCC) (left) and Gross Primary Production (right) for the results of each model simulation. NEE from the model is compared to the GCP 2017 and INV data set (a and c). GPP is compared to the MTE data-data driven estimate of Jung et al., (2011) (b and d).



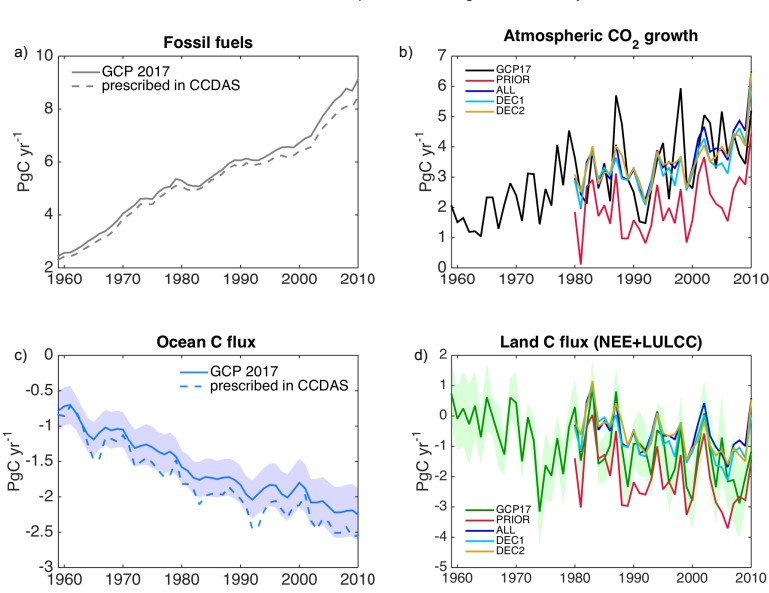

Figure 8 – Time series of the annual mean of the major components of the C cycle used as background fluxes in CCDAS compared to those from the GCP 2017. The atm. $CO_2$ growth from the model output is the result of the sum of fossil fuel, ocean, and land C fluxes. The blue shadow in the ocean C sink of the GCP 2017 data is the standard deviation of the mean sink from the models that contributed to the GCP. The land C flux is the total NEE with contribution of the flux due to LULCC. The green shadow area is the standard deviation of the mean land C flux from the terrestrial models that contributed to the GCP.



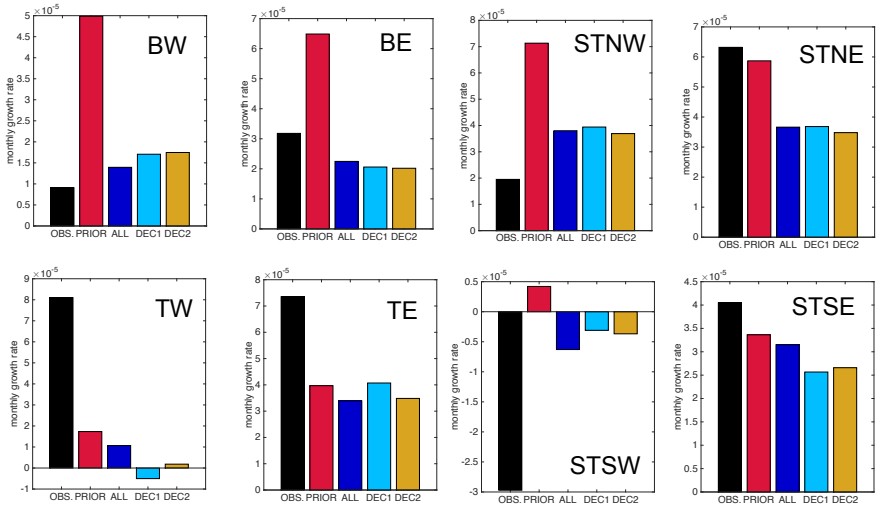

Figure 9 – Mean monthly growth rate of FAPAR for 1982-2006 on each analyzed geographical region for the satellite observations and results of PRIOR and the posterior experiments.

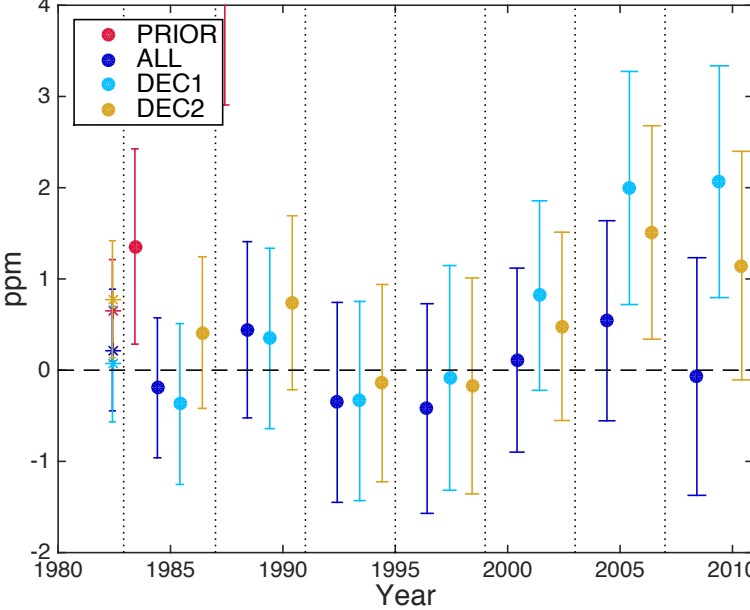

Figure 10 – Time series of the four-year mean of the atm. $CO_2$ anomaly to the observations for each model experiment and for all the stations. The y-axis is limited to the results in the posterior experiments. The error bar indicates $+/-$ 1 standard deviation of the four-year mean of the differences to the observations. The first marker in the time series (in asteric) is the single value for 1982.





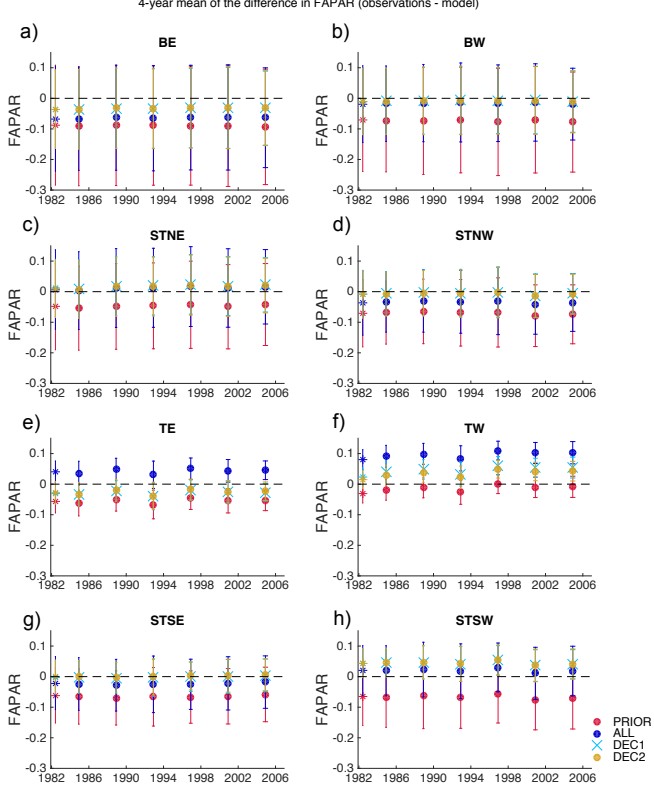

Figure 11 – Time series of the four-year mean of the FAPAR anomaly to the satellite data for each model experiment in six selected model pixels. The error bar indicates the +/− 1 standard deviation of the four-year differences. The first marker (in asteric) in the time series is the single value for 1982.





**Appendix**

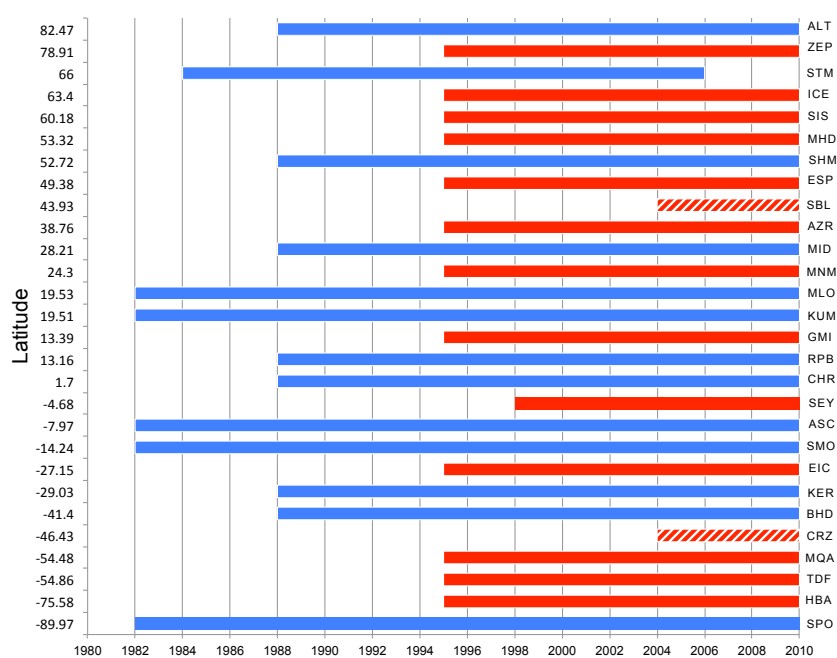

Figure A1 – Data availability and latitudinal location of the 28 stations where the long-term flask measurements of atmospheric $CO_2$ mole fractions were taken for assimilation in CCDAS. ALL experiment used all the stations of the time series (blue and red bars) (1980-2010); DEC1 used data only from stations with blue bars (1980-1990), and DEC2 used also the data in the stations with red bars (1990-2000) (except stations SBL and CRZ marked with patterned bar).



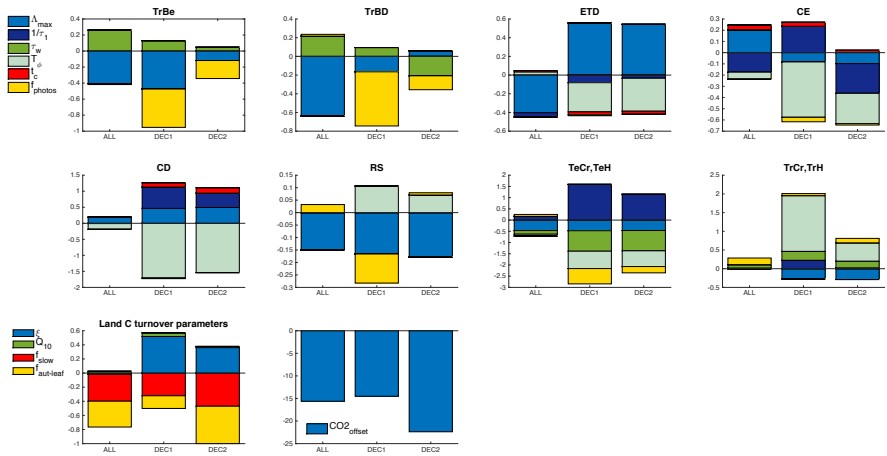

Figure A2 – Final value for each parameter $p$ at the end of the assimilation experiments, normalized to the prior value ($p_{pr}$), i.e. ($p/p_{pr}$)-1. This is shown for each model plant functional type and globally for the land C turnover parameters.

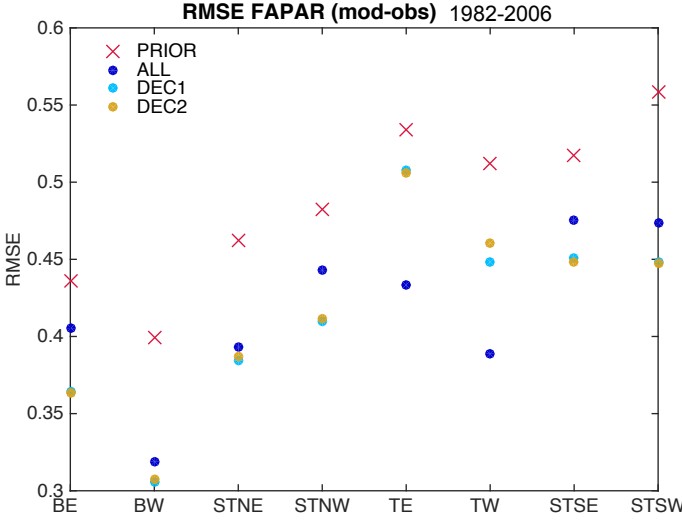

Figure A3 – RMSE for FAPAR from the model results and observations for the period 1982-2006 and for different regions.





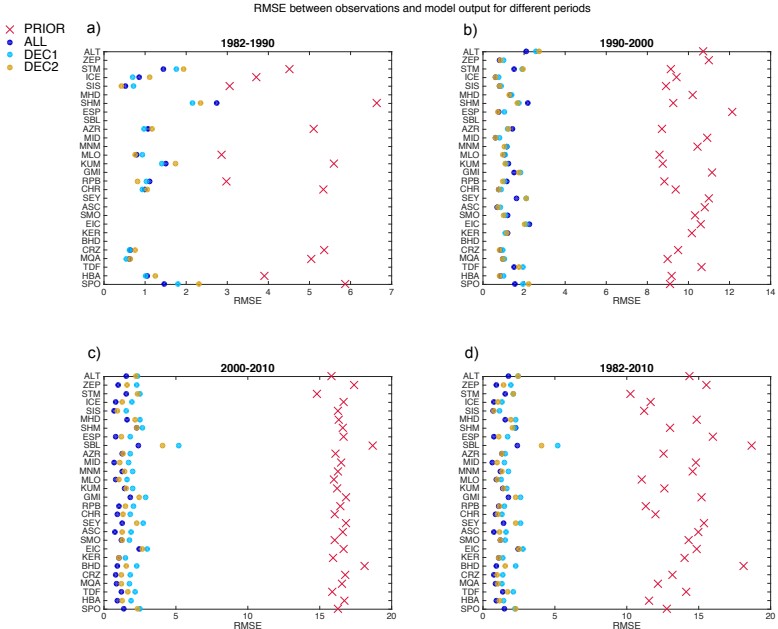

Figure A4 – RMSE for different periods between $CO_2$ atm. concentrations from
measurements and model results for the different assimilation experiments for each of
the station.