# Peer review of "Three decades of simulated global terrestrial carbon fluxes from a data assimilation system confronted with different periods of observations."

_Biogeosciences, 2018_

## Referee Comment (RC1) · Anonymous Referee #1 · 20 Feb 2019

This paper develops an analysis of the global land C fluxes for three decades. The aim of the paper is to analyse the performance of a data assimilation system, over a historical period. The paper misses clear science questions that would broaden the interest of the study. I lack a clear understanding of the novelty of this work. The conclusions focus on the result that a single decade of data leads to similar results to 3 decades of assimilation, but I am not clear how robust this result is, and how significant it is.

I suggest the authors focus more clearly on the novelty of their experimental results. It

would be helpful to focus on science questions that are more broadly relevant to the C cycle science community. It might be helpful to add experiments assimilating just FAPAR or just CO2, and test the outputs against the other (withheld) observational dataset. This experiment might indicate whether the model effectively couples canopy processes with atmospheric concentrations.

The abstract is too long, it needs to be reduced to 300 words and focused on the key outcome.

Writing style is clunky, missing words – for instance the opening sentence is poorly comprehensible: "The observed contemporary in atmospheric CO2 is driven by anthropogenic emissions from fossil fuels and land-use change". I suggest the authors work together to improve the English, reduce sentence lengths, and shorten the more bloated paragraphs.

Methods: The global grid is very coarse  $(8x10^{\circ})$  – what are the implications? Why mix such a large spatial grid with such fine temporal scales? Is this valid or required? Why not have a grid that matches TM3? Explain what is meant by 'each iteration' (L. 139). It is unclear how FAPAR data are generated from equation 2, which seems to generate NDVI estimates. We require details on the FAPAR observation operator – I could not find any. Why are there no FAPAR data after 2006? (I. 278) What are the implications of not including fire emissions? Why were fire products such as GFED not used to provide this input? I would like to know more about the process to determine which parameters were selected for optimisation? These seems to mostly phenological variables which will link to FAPAR. I would expect to see other parameters related to C turnover, e.g. mortality rates, decomposition rates.

Results: What are the substantial changes in tropical LAI (I. 306) – how do these match in situ measurements? What is the increase in R2 (I, 314) – please report in the text. The key result seems to be shifts in the timing of CO2 exchanges – it would be interesting to focus more on the shifts in model process representation (parameters)
required to allow these changes. I do not find Figure A2 very helpful in this regard. The large drop in GPP in the posteriors is significant – what process is this traced to in the parameter adjustment? I did not find fig 4 helpful in regard to identifying improvements in IAV modelling – it would help to have some statistics to support the statements here (l. 461).

**Discussion**

"the mismatch between observations and model output is small, and thus of little concern". This statement needs to be more rigorous – how is 'small' determined? What is the threshold for concern? The lack of tropical IAV suggests too weak an ENSO response – were any relevant parameters included in the CCDAS that would have allowed identification of drought response? Some of the discussion seems circular – that using FAPAR data in the assimilation improves modelling of FAPAR seasonality. It would be good to clarify the text to the explicit calibration and validation results to strengthen this section. It seems this result has already been identified in an earlier CCDAS publication, so it is not clear what is novel here.

The authors identify problems "results from the structural dependence of the MPI-CCDAS on few, globally applicable PFT-level parameters, and challenges in using the spatial mixed signal at the model resolution to infer PFT-specific parameters." It would help to develop these ideas some more – do we expect these issues to specifically affect the current analysis and in what ways?

Table 3. The posteriors suggest a Ra:GPP ratio of  ${\sim}65\%$  - it would be useful to discuss this value which seems high for a global estimate.

The very large reduction in soil C stocks from the prior needs further discussion – JS-BACH was spun up to steady state for the prior, so I am not clear how the experiments generated 50% drops in this value. How far is the model from steady state with such a reduction in soil C?
We hear again that a result here repeats an earlier CCDAs result (I. 660), which reduces the novelty of the analysis.

Other comments:

Figures – there are too many figures, some of which are of low value, and this distracts from the key message of the paper. Some of the figures in the appendix are referred to several times, so why are they not in the main text (replacing those referenced less).

L. 59 . Citation needed for this 5.6% value

L. 297. The zone between 20-60° is not well described as "sub-tropical"

Table 1: Add row numbers

---

## Referee Comment (RC2) · Anonymous Referee #2 · 6 Mar 2019

This is an interesting and useful paper, albeit of more technical than scientific interest. There are a number of factors that reduce the scientific impact of the paper, while focusing more on the interaction of observations with a model of this type when used in assimilation mode. The reanalysis is limited by a number of factors, the very low spatial resolution dictated (I suppose) by the resolution of the atmospheric inverse model, the limited data fields assimilated (just carbon fluxes and FAPAR) and the lack of potentially important processes, such as fire. The author's assessment of model skill is ambivalent, they point to low errors in some places, while noting that the El

Nino cycle is not well-captured, a time scale that others have argued provides a critical clue to climate sensitivity (eg Cox et al). By contrast, the advanced methods used and the useful assessment of the impact of the duration of the assimilation experiment, as well as other technical innovations provides a useful update to their prior paper, as the scientific conclusions are overlapping. As assimilation becomes more prevalent, and as data records lengthen (for this study, of a 30-year time scale these really are the most relevant global fields) with SIF, radar-constrained biomass, and water variables such as vegetation optical depth becoming available for > 10 years, this paper provides encouraging news about the utility and impact of records of decadal length. I'd suggest rewriting the paper modestly to emphasize the lessons learned about the impact of assimilation, and the time horizons, and placing less emphasis on the carbon cycle results, especially as the authors note (and correctly) the conclusions broadly overlap their earlier paper. I note that papets of the paper are awkwardly written and could use a careful edit, and there are a lot of figures—I found them helpful in reviewing the paper but several of the figures could clearly be moved to supplemental material.

---

## Author Comment (AC1) · 5 Apr 2019

We appreciate the comments from two anonymous reviews. We answer below in blue text to each comment listed in black adding the explicit action taken in a revised manuscript.

**Reviewer #1**

This paper develops an analysis of the global land C fluxes for three decades. The aim of the paper is to analyse the performance of a data assimilation system, over a historical period. The paper misses clear science questions that would broaden the interest of the study. I lack a clear understanding of the novelty of this work. The conclusions focus on the result that a single decade of data leads to similar results to 3 decades of assimilation, but I am not clear how robust this result is, and how significant it is. I suggest the authors focus more clearly on the novelty of their experimental results. It would be helpful to focus on science questions that are more broadly relevant to the C cycle science community.

We appreciate the reviewer's concerns and are grateful for the comments, which have helped us to formulate the manuscript more clearly. We present three decades of land C flux reanalysis with the aim to better understand the ability of MPI-CCDAS to make decadal projections of the land C cycle. Our approach was to assimilate different time-periods of data to understand the effect of data selection, but also the effect of withholding more recent information. In this way, we can assess the projection of carbon fluxes for the more recent period not simply against the withheld observations, but also against a model run, which has been constrained by this data, in order to disentangle model limitations (i.e. in reproducing an observation that the model was informed about through data assimilation) from prognostic uncertainty (i.e. in failing to reproduce an observation that it can reproduce when given this information). To our knowledge, this is a novel design and focus. Previous studies looking at the prognostic capacity of a CCDAS only looked at short timescales after the assimilation (Scholze et al. 2007; Schürmann et al. 2016), or in the case of Rayner et al. 2011, the authors used a much simpler model ignoring the interacting effects of water, energy and phenology on the carbon cycle predictions. In the revised manuscript, we will be more explicit and clear in the scientific contribution of our results that are relevant to the carbon cycle scientific community.

Our results demonstrate that it is not necessary to ingest more than one decade of observations to improve important features of the global carbon cycle over the following 1-2 decades, in particular the long-term trend and seasonal amplitude of atmospheric $CO_2$ concentrations at station level, as well as in the long-term trend, phenological seasonality and interannual variability of FAPAR. These results provide an insight into the amount of data that is necessary in data assimilation systems to improve the representation of the global carbon cycle components. This information might be decisive for opening the possibility of including newly measured data of other global indicators, such as SIF, that only exist for periods of less than a decade.

It might be helpful to add experiments assimilating just FAPAR or just CO2, and test the outputs against the other (withheld) observational dataset. This experiment might indicate whether the model effectively couples canopy processes with atmospheric concentrations.

We appreciate the suggestion of the reviewer regarding the experiment of only assimilating one of the observational data sets at a time. However, we did not want to repeat this experiment because this has been previously done with the same system in

Schürmann et al. 2016. In that work, 5 years of observational data (2005 – 2009) were assimilated in three independent experiments: only FAPAR, only atmospheric $CO_2$ concentrations and the two data sets simultaneously. The results showed that with this time period it was sufficient to demonstrate, in a nutshell, that when assimilating only FAPAR, the average growing season and vegetation seasonality (indicated by FAPAR) was considerably improved in the northern boreal areas. When atmospheric $CO_2$ concentrations were the only assimilated observations, the global gross and net carbon fluxes were overall improved, but the GPP in the tropics was significantly reduced when compared to the GPP after the FAPAR-only assimilation.

While we agree with the reviewer that in principle it would have been interesting to repeat these experiments to analyze their effect on long-term trends, this would require too much computational time to be included in this study and it was not the central focus of this work.

The abstract is too long, it needs to be reduced to 300 words and focused on the key outcome.

We will significantly shorten the abstract and remain under 300 words including only the key outcome regarding using different periods in the observational data during the assimilation.

Writing style is clunky, missing words – for instance the opening sentence is poorly comprehensible: "The observed contemporary in atmospheric CO2 is driven by anthropogenic emissions from fossil fuels and land-use change". I suggest the authors work together to improve the English, reduce sentence lengths, and shorten the more bloated paragraphs.

We realized that we missed the word "increase" in the opening sentence and appreciate the reviewer for pointing this out. The sentence should read: "The observed contemporary **increase** in atmospheric $CO_2$ …", we apologize for this. Besides this correction, the revised manuscript will undergo a thorough revision for the English language with the aim of shortening sentences and paragraphs.

Methods: The global grid is very coarse (8x10_) – what are the implications?

A previous work with the dynamic global vegetation model LPG-DGVM (Müller and Lucht, 2007), demonstrated that the effect of increasing the spatial resolution did not have a strong effect on the simulated regional and global carbon fluxes, whereas temporal dynamics are unaffected. To our knowledge, the effects on the simulated C fluxes after assimilation due to changes in the horizontal spatial resolution of the model grid cells solely, has not yet been assessed with current data assimilation systems. One reason for this is that the computational cost of such an approach is prohibitive. A study using ORCHIDAS (Peylin et al., 2016) suggested that the level of complexity of the ecosystem model in a data assimilation system, including its spatial resolution, does not guarantee improvements in the optimization of C fluxes. Such a conclusion is likely valid for the assimilation of atmospheric observations, but does decrease the ability of the CCDAS to adequately constrain phenology parameters given the mixed phenology in these large pixels. We would like to point out, however, that increasing the spatial resolution to a degree that would allow for a clean identification of PFT-wise parameters, exceeds the resolution of many state-of-the-art forward terrestrial biosphere models. We will add this discussion in the revised manuscript.

Why mix such a large spatial grid with such fine temporal scales? Is this valid or required? Why not have a grid that matches TM3?

We are unsure on the meaning of the question by the reviewer, because it combines both spatial and temporal scales. In the CCDAS, we integrate the simulated daily net $CO_2$ fluxes to monthly scale and transport them with the Jacobian representation of TM3. This approach allows us to account for the non-linear impact of weather anomalies on the surface fluxes, but removes the impact of synoptic atmospheric transport variability on the simulated seasonal and long-term dynamics of atmospheric $CO_2$ at the monitoring stations. We will improve this explanation in the revised manuscript.

We would have liked to use MPI-CCDAS on the same grid as TM3; however, the land-surface model is currently numerically too expensive to allow these higher resolutions.

Explain what is meant by 'each iteration' (L. 139).

The term "each iteration" refers to every cycle when the model re-calculates the cost function for the difference between the model parameters and the observational constraint. We will add this explicitly to the revised ms for clarification.

It is unclear how FAPAR data are generated from equation 2, which seems to generate NDVI estimates. We require details on the FAPAR observation operator – I could not find any.

We apologize for the error in Eq. 2, which should have read:

"We therefore used as FAPAR proxy the Global Inventory Monitoring and Modeling System (GIMMS) NDVI product for the period 1982 to 2006 (Tucker et al., 2005), and merged it with the TIP-FAPAR product to provide a longer record of vegetation greenness. The maximum and minimum NDVI values were rescaled at the pixel level to coincide with those from the TIP-FAPAR for the overlapping time periods (i.e. 2003 to 2006) following:

$$\text{FAPAR}_{mod} = \frac{\text{NDVI} - \text{NDVI}_{min,x}}{\text{NDVI}_{max,x} - \text{NDVI}_{min,x}} \times \left(\text{TIP}_{max,x} - \text{TIP}_{min,x}\right) + \text{TIP}_{min,x} \qquad (2)$$

where x is the period 2003 to 2006 for each data set, $\text{FAPAR}_{min}$ and $\text{FAPAR}_{max}$ are from the NDVI product, NDVI is the full product from 1982 to 2006, and $\text{TIP}_{max}$ and $\text{TIP}_{min}$ are from the TIP-FAPAR product. In this way, the merged product maintains the maximum and minimum values from TIP-FAPAR while preserving the temporal dynamics of NDVI. Due to a technical failure in the CCDAS, the final $\text{FAPAR}_{mod}$ product spans only from 1982 to 2006 and the last four years from TIP-FAPAR product (2007-2010) were not included."

Why are there no FAPAR data after 2006? (l. 278)

Unfortunately, this is an error during the assimilation procedure. Due to a technical fault, the CCDAS did not consider the remaining four years of data from the original TIP-FAPAR time-series as planned. We discovered this issue only during the post-processing phase, and we wrote it in the final sentence of the paragraph above. However, we believe that our results are still valid, because the main information gain of the CCDAS in terms of phenology stems from the seasonal cycle, with little effect on the overall trends between the three assimilation experiments with different time

periods. This comment will be added also in the revised manuscript.

What are the implications of not including fire emissions? Why were fire products such as GFED not used to provide this input?
Omitting fire fluxes may impair the ability of the MPI-CCDAS to correctly infer the atmospheric growth rate of $CO_2$ in years with strong contribution of a fire flux, such as the 1998 El Niño. It was not possible to use GFED-like products in this particular assimilation experiment, because for example GFED4 data do not exist prior to 1997, whereas our inversion started in 1982. Adding them only starting in 1997 would have biased the assimilation procedure. In the discussion manuscript we added already the lack of this process as part of the limitations of the MPI-CCDAS in the discussion section.

I would like to know more about the process to determine which parameters were selected for optimisation? These seems to mostly phenological variables which will link to FAPAR. I would expect to see other parameters related to C turnover, e.g. mortality rates, decomposition rates.
As described in Schürmann et al. 2016, the choices of parameters were done by an extensive parameter sensitivity study with a large set of MPI-CCDAS model parameters for a wide range of biomes. We retained parameters that had a strong effect of the simulated carbon and water fluxes as well as phenology. In Table 1, we listed the parameters selected for the optimization. While the majority are indeed linked to phenology, we considered also parameters linked to photosynthesis and global parameters that control the land carbon turnover. Those are the last four parameters listed in Table 1: for the heterotrophic respiration the temperature sensitivity to respiration ($Q_{10}$) and a multiplier for initial slow pool ($f_{slow}$) to account for non-steady state conditions at the beginning of the assimilation; for the autotrophic respiration, the leaf fraction of maintenance respiration, and finally an initial atmospheric carbon concentration.

**Results:**
What are the substantial changes in tropical LAI (l. 306) – how do these match in situ measurements?
As shown in Table 1, the maximum LAI value is one of the optimization parameters and it was prescribed for each PFT. In the case of the tropical evergreen and deciduous trees, this equals to 7 $m^2$ $m^{-2}$. To provide a numerical context to the LAI changes, we summarize in the table below the mean and maximum LAI values per regions of Fig. 1 for each experiment, and we also show in the figure below, the average maximum LAI for each experiment for the period 1980-2010.

| Region | PRIOR (mean / max) ($m^2$ $m^{-2}$) | ALL (mean / max) ($m^2$ $m^{-2}$) | DEC1 (mean / max) ($m^2$ $m^{-2}$) | DEC2 (mean / max) ($m^2$ $m^{-2}$) |
|---|---|---|---|---|
| BE | 0.61 / 2.29 | 0.60 / 1.94 | 0.70 / 2.42 | 0.69 / 2.42 |
| BW | 0.31 / 1.62 | 0.30 / 1.44 | 0.35 / 2.01 | 0.35 / 2.02 |
| TNE | 1.28 / 4.28 | 1.17 / 3.33 | 1.31 / 3.49 | 1.32 / 3.79 |
| TNW | 1.26 / 3.11 | 1.15 / 2.84 | 1.30 / 3.23 | 1.30 / 3.21 |
| TE | 1.62 / 3.27 | 1.30 / 2.43 | 1.63 / 3.20 | 1.67 / 3.33 |
| **TW** | **2.21 / 3.17** | **1.68 / 2.27** | **2.00 / 2.89** | **2.08 / 3.00** |
| TSE | 1.54 / 2.72 | 1.43 / 2.51 | 1.86 / 2.77 | 1.83 / 2.68 |
| TSW | 2.42 / 3.69 | 2.04 / 2.71 | 2.38 / 3.47 | 2.43 / 3.66 |

[Figure]

Max LAI [m^2/m^2] (1980-2010)

When we compare the LAI mean values between the experiments and the PRIOR results (Fig. 3 in the discussion manuscript), we observe that the largest change in LAI values was in the tropical west area (TW) comprising Brazil, with a decrease in LAI values of up to 24 % in the ALL experiment with respect to the PRIOR, as a response of the maximum LAI decay in the tropical evergreen PFT (visible in the figure above). We argue that this decrease is a response of a global compensating effect to heterotrophic respiration, leading to the lower GPP tropical value.

Ground based observations in the tropical Amazon-Savanna transition forest have been reported with an annual mean LAI value for the total canopy between 2005 and 2008 of $7.4\pm0.6$ $m^2$ $m^{-2}$, and for the seasonal flooded forest a value of $3.4\pm0.1$ $m^2$ $m^{-2}$. For the remote sensing data from MODIS, the reported values are $6.2\pm0.2$ $m^2$ $m^{-2}$ and $5.8\pm0.3$ $m^2$ $m^{-2}$, respectively (Biudes et al., 2014).

In the eastern Amazon forest, the remote sensing-based LAI has been reported as 6.2 $m^2$ $m^{-2}$ from LiDAR, and 4.8 $m^2$ $m^{-2}$ with a low end of 2.0 $m^2$ $m^{-2}$ from MODIS (Qu et al., 2011). The maximum LAI values from our model results before and after the assimilation (see table above) fall within the values from MODIS and LiDAR. However, this comparison is robust because of the spatial resolution of the different methods: a coarse model grid cell resolution vs. the resolution of ground-based measurements and the resolution of the remote sensing pixels (50x50 m for ground-based and LiDAR data, and 463 x 463 m for MODIS).

As mentioned below, Figure 3 will be moved to the supplement for the sake of keeping the revised manuscript more focused on the main aim, and because the reference to the LAI results were done only sporadically through the manuscript. We will also add (at the editors discretion) another figure in the supplement to show the differences between experiments for the maximum LAI of the mean values.

[Figure]

Max LAI [m^2/m^2] (1980-2010)

What is the increase in R2 (l, 314) – please report in the text.

The $R^2$ values between the FAPAR observations and model results are: 0.1638 for PRIOR, 0.1984 for ALL, 0.3412 for DEC1 and 0.3402 for DEC2. The values are given in Lines 318-323. We will improve these paragraphs to make clearer that the values are given in those lines by moving the values to the lines above.

The key result seems to be shifts in the timing of CO2 exchanges – it would be interesting to focus more on the shifts in model process representation (parameters) required to allow these changes. I do not find Figure A2 very helpful in this regard.

We appreciate the comment from the reviewer, however we believe that an interpretation of the results at the level of changes in parameters only adds an incomplete picture to the analysis and it is difficult to conclude the effect of the changes. For this reason, we only focused on the relative changes summarized in Fig. A2. We will add a section in the Appendix regarding the assimilation performance where we discuss in more detail the response of each parameter.

The large drop in GPP in the posteriors is significant – what process is this traced to in the parameter adjustment?

This results primarily from a reduction in tropical leaf area index (see Figure 3 in discussion manuscript that will be moved to the supplement for the revised version, and also in the comment above), as well as the drop in the photosynthetic capacity (see larger change in Fig. A2 for parameter $f_{photos}$ in the tropical evergreen and deciduous PFTs) after the assimilation. In our results, we will explicitly add these observations in the appendix part of the assimilation performance mentioned above.

I did not find fig 4 helpful in regard to identifying improvements in IAV modelling – it would help to have some statistics to support the statements here (l. 461).

In the revised manuscript, we will present summary statistics (RMSE, $R^2$) in the text, and will move the Figure to the supplementary material.

**Discussion**

"the mismatch between observations and model output is small, and thus of little concern". This statement needs to be more rigorous – how is 'small' determined? What is the threshold for concern? The lack of tropical IAV suggests too weak an ENSO response – were any relevant parameters included in the CCDAS that would have allowed identification of drought response? Some of the discussion seems circular – that using FAPAR data in the assimilation improves modelling of FAPAR seasonality.

We will clarify this statement a) to note that this corresponds to the IAV of FAPAR, b) that the observed signal is small compared to seasonal variations, and c) the retrieval error or the FAPAR product, which as a global average corresponds to ±0.2088 (relative units, Schürmann et al. 2016). The assimilation procedure allows changes in the phenology response to water stress ($\tau_w$). However, the assimilation tended to decrease rather than increase the drought sensitivity of tropical phenology given the entire spatially explicit FAPAR time series, and therefore did not allow to capture these excursions (assuming that they are actually driven by drought related changes in LAI).

It would be good to clarify the text to the explicit calibration and validation results to strengthen this section. It seems this result has already been identified in an earlier CCDAS publication, so it is not clear what is novel here.

In the revised manuscript, we will clarify the text and reduce redundancy to the earlier publication. However, since we are using new data and run a different set-up, we believe that it is important to establish the baseline performance of the CCDAS before looking into the novel results of long-term trends. We will condense this section to the absolute necessary to give more space to the novel results (see also our response to the selection of Figures below).

The authors identify problems "results from the structural dependence of the MPICCDAS on few, globally applicable PFT-level parameters, and challenges in using the spatial mixed signal at the model resolution to infer PFT-specific parameters." It would help to develop these ideas some more – do we expect these issues to specifically affect the current analysis and in what ways?

In the revised manuscript, we will expand this discussion. Firstly, this relates to the problem mentioned by the reviewer above with respect to the impact of coarse spatial resolution. We believe that aggregating the remote sensing data into PFT-specific

classes per pixel from a high-resolution grid would allow reducing the problems in the identification of phenological parameters. Secondly, although some of the phenological parameters adapt to mean growing season temperature, some of the thresholds are globally applicable, which causes mainly a problem for temperate grasslands, which cover a wide climatological range. Finding an appropriate means to cluster grasslands into more spatially refined classes would further reduce the errors of the MPI-CCDAS to simulating boreal, temperate and tropical phenology. Finally, some of the global parameters (such as $f_{aut\_leaf}$ and $f_{slow}$) imply that improvements of modeled fluxes in the boreal regions directly affect fluxes in the tropics, inducing parameter changes to compensate for the altered C fluxes. Such dependence is typical in biosphere models, but may not be ecologically and eco-physiologically correct. Defining these parameters per PFT would reduce such a problem. However, any of these changes would inflate the inverse problem to be solved, therefore increasing computational costs and would not necessarily reduce overall uncertainty (equifinality).

Table 3. The posteriors suggest a Ra:GPP ratio of _65% - it would be useful to discuss this value which seems high for a global estimate.
We believe that this features is a result of the fact that net primary production itself is not well constrained from the atmospheric record. We suspect that two factors contribute to the low NPP:GPP ratio: i) the observed fast coupling between GPP and both autotrophic and heterotrophic respiration, which cannot be reproduced by a state-of-the-art first-order-decay soil carbon turnover model. Since autotrophic respiration in MPI-CCDAS is directly coupled to GPP, increasing the fraction of GPP partitioned to it increases the seasonal cycle of ecosystem respiration; ii) Increasing Ra reduces the net land carbon uptake, and may mask changes in vegetation carbon turnover, which were excluded from the analysis, because their effect on carbon storage was much lower than that of changing $f_{aut\_leaf}$. We note that accounting for the vegetation carbon turnover parameters without any further constraint on NPP would likely not have increased the confidence in the CCDAS outcome because of equifinality.

The very large reduction in soil C stocks from the prior needs further discussion – JSBACH was spun up to steady state for the prior, so I am not clear how the experiments generated 50% drops in this value. How far is the model from steady state with such a reduction in soil C?
The model was spun-up initially until the soil carbon pools reached equilibrium considering pre-industrial forcing. However this new "initial state" for the model is not on steady state when considering climate variability, hence to compensate this, the CCDAS creates an artificial sink of C, leading to a reduction in the soil C stocks, in order to reduce the respiration. Unfortunately, this is unavoidable and is rather a model effect to compensate by contemporary climate changes.

We hear again that a result here repeats an earlier CCDAs result (l. 660), which reduces the novelty of the analysis.
To investigate the mechanisms that influence the patterns observed in the simulated global GPP or NEP after the assimilation is out of the scope of the presented manuscript. The reference to the work of Schürmann et al. 2016 in this line, as well as in previous others throughout the ms, serves as a point of comparison to previous results with the same model but obtained under a different experimental design (5 years only of assimilation), which also contributes to set the preceding performance of

the current set up (see comment above).

Other comments:
Figures – there are too many figures, some of which are of low value, and this distracts from the key message of the paper. Some of the figures in the appendix are referred to several times, so why are they not in the main text (replacing those referenced less).
With the aim of shortening the manuscript and avoid distraction to the main findings, we will remove some figures for the revised version. Specifically we will remove those figures that are less referred to in the main manuscript such as: Fig. 2, on the experimental design and mentioned only once in the main text, it will be moved to appendix. Figures 3 and 4 will be moved to the supplement: Fig. 3, showing the spatial distribution of mean LAI before and after the assimilation, but the manuscript does not focus on the specific changes on LAI after the assimilation, instead $R^2$ values are given in Table 2. Fig. 4 shows the interannual variability of FAPAR for the different sub-regions is only mentioned briefly in the results and will be moved to the supplement.
As from figures from the appendix Figures A3 and A4 were mentioned more frequently hence they would be moved to the main text.

L. 59 . Citation needed for this 5.6% value
The reference for this value is LeQueré et al., 2018 which is cited in the lines below.

L. 297. The zone between 20-60_ is not well described as "sub-tropical"
We will refer to this range of latitudes as north and south temperate zones (TNW and TNE for west and east northern hemisphere, and TSW and STE for west and east southern hemisphere) in the revised manuscript.

Table 1: Add row numbers
Ok, in a first column we will add row numbers.

**References listed in this response.**

Biudes, M. S., Machado, N. G., de Morais Danelichen, V. H., Caldas Souza, M., Vourlitis, G., and Nogeuira, J. d. S.: Ground and remote sensing-based mesurements of leaf area index in a transitional forest and seasonal flooded forest in Brazil, International Journal of Biometeorology, 58, 1181-1193, 2014, 10.007/s00484-013-0713-4.

Lasslop, G., Thonicke, K., and Kloster, S.: SPITFIRE within the MPI Earth system model: model development and evaluation, Journal of Advances in Modeling Earth Systems, 6, 740-755, 2014, 10.1002/2013MS000284.

Müller, C. and Lucht, W.: Robustness of terrestrial carbon and water cycle simulations against variations in spatial resolution, Journal of Geophysical Research, 112, D06105, 2007, 10.1029/2006JD007875.

Peylin, P., Bacour, C., MacBean, N., Leonard, S., Rayner, P., Kuppel, S., Koffi, E., Kane, A., Maignan, F., Chevallier, F., Ciais, P., and Prunet, P.: A new stepwise carbon cycle data assimilation system using multiple data streams to constrain the

simulated land surface carbon cycle, Geoscientific Model Development, 9, 3321-3346, 2016, 10.5194/gmd-9-3321-2016.

Qu, Y., Shaker, A., Silva, C. A., Klauberg, C., and Rangel Pinagé, E.: Remote sensing of Leaf Area Index from LiDAR height percentile metrics and comparison with MODIS product in a selectivley logged tropical forest area in Eastern Amazonia, Remote Sensing, 10, 1-23, 2011, 10.3390/rs10060970.

Tucker, C. J., Pinzon, J. E., Brown, M. E., Slayback, D. A., Pak, E. W., Mahoney, R., Vermote, E. F., and El Saleous, N.: An extended AVHRR 8-km NDVI dataset compatible with MODIS and SPOT vegetations NDVI data, International Journal of Remote Sensing, 26, 4485-4498, 2005, 10.1080/01431160500168686.

---

## Author Comment (AC2) · 5 Apr 2019

We appreciate the comments from two anonymous reviews. We answer below in blue text to each comment listed in black adding the explicit action taken in a revised manuscript.

**Reviewer #2**

This is an interesting and useful paper, albeit of more technical than scientific interest. There are a number of factors that reduce the scientific impact of the paper, while focusing more on the interaction of observations with a model of this type when used in assimilation mode.

The reanalysis is limited by a number of factors, the very low spatial resolution dictated (I suppose) by the resolution of the atmospheric inverse model, the limited data fields assimilated (just carbon fluxes and FAPAR) and the lack of potentially important processes, such as fire.

Reviewer #2 rightly points out the limitations to our study. However, we think that it still offers relevant insights despite its limitations, which have also been pointed out by reviewer #2. We will revise the manuscript to discuss the limitations of the current study more in detail, while at the same time, focusing on the main story of the manuscript: the use and effect of long-term data sets for assimilation and the question on how long the improved model/data agreement can last. We hope that this study will inspire the future use of CCDAS systems to integrate further data streams (such as SIF or VOD), for which a CCDAS is uniquely suited given it's ability to use data at different resolutions and for different time-periods. We will discuss this more clearly in the revised manuscript.

As in our response to reviewer #1, the spatial resolution is indeed dictated by the computational setup. Increasing resolution would of course allow for a better integration of remote sensing data as well as the current sub-grid scale variability in climate. However, as has been pointed out before (Müller and Lucht, 2007; Peylin et al., 2016), have suggested that increased resolution would not necessarily have a strong effect on the overall performance of the model against global carbon cycle observations.

As in any model study, MPI-CCDAS does not include all processes. As noted in our response to reviewer #1, using data sets to account for this flux is not possible given the lack of data before 1997. While there is a fire module in MPI-CCDAS (Lasslop et al., 2014), it has been identified a number of issues with that module that would need to be addressed, and the effect of these issues on the spatio-temporal dynamics of the land carbon balance would need to be clarified before it is possible to include it into these long-term and computationally intensive MPI-CCDAS runs. We agree that the addition of disturbance processes due to fires is an interesting aspect for future MPI-CCDAS developments and may contribute to an improved representation of the interannual variability. However, we note that some of the major fluxes (deforestation, peatland fires) are not considered by this, and many other, fire models. In the revised manuscript, we will include a paragraph containing the potential implications to our results of not having explicitly included fire emissions in our experiments.

The author's assessment of model skill is ambivalent, they point to low errors in some places, while noting that the El Nino cycle is not well-captured, a time scale that others have argued provides a critical clue to climate sensitivity (eg Cox et al).

In the revised manuscript we will clarify this issue, which is probably at the core of many model-data inter-comparison studies: the fact that cost functions measure the absolute misfit of a model versus data, and are not necessarily sensitive to important aggregate system properties such as the response of the tropical carbon cycle to ENSO. Our assimilation study clearly exposes this problem, as the assimilation stops without such known variability considered and this is because in the cost-function the benefit of matching this interannual variability is not strongly weighted. In the revised manuscript, we will discuss the implications and potential ways around such problems.

By contrast, the advanced methods used and the useful assessment of the impact of the duration of the assimilation experiment, as well as other technical innovations provides a useful update to their prior paper, as the scientific conclusions are overlapping. As assimilation becomes more prevalent, and as data records lengthen (for this study, of a 30-year time scale these really are the most relevant global fields) with SIF, radar-constrained biomass, and water variables such as vegetation optical depth becoming available for > 10 years, this paper provides encouraging news about the utility and impact of records of decadal length.

We appreciate the comments and support from the reviewer to this manuscript and for valuing the scientific contribution of our work.

I'd suggest rewriting the paper modestly to emphasize the lessons learned about the impact of assimilation, and the time horizons, and placing less emphasis on the carbon cycle results, especially as the authors note (and correctly) the conclusions broadly overlap their earlier paper. I note that papets of the paper are awkwardly written and could use a careful edit, and there are a lot of figures I found them helpful in reviewing the paper but several of the figures could clearly be moved to supplemental material.

We believe that it is important to demonstrate that the carbon cycle results of a 30 years and a 5 years experiment (as in Schürmann et al. 2016) are broadly comparable to set the stage for the impact of the different time horizons. However, we agree with both reviewers that in the previous version this obscured the key innovation of the study and we will therefore revise the manuscript to make this clearer. More concretely, we will place much of the evaluation material and associated text to the supplementary material, and instead give more space to the results and discussions of the time horizons. Specifically: Fig. 2, on the experimental design and mentioned only once in the main text, it will be moved to appendix. Figures 3 and 4 will be moved to the supplement: Fig. 3, showing the spatial distribution of LAI before and after the assimilation, but the manuscript does not focus on the specific changes on LAI after the assimilation, instead $R^2$ values are given in Table 2. Fig. 4 shows the interannual variability of FAPAR for the different sub-regions is only mentioned briefly in the results and will be moved to the supplement. As from figures from the appendix Figures A3 and A4 were mentioned more frequently hence they would be moved to the main text.

With the aim of delivering a clearer message in our manuscript, the revised version will undergo a thorough English revision before re-submission.

**References listed in this response.**

Lasslop, G., Thonicke, K., and Kloster, S.: SPITFIRE within the MPI Earth system model: model development and evaluation, Journal of Advances in Modeling Earth Systems, 6, 740-755, 2014, 10.1002/2013MS000284.

Müller, C. and Lucht, W.: Robustness of terrestrial carbon and water cycle simulations against variations in spatial resolution, Journal of Geophysical Research, 112, D06105, 2007, 10.1029/2006JD007875.

Peylin, P., Bacour, C., MacBean, N., Leonard, S., Rayner, P., Kuppel, S., Koffi, E., Kane, A., Maignan, F., Chevallier, F., Ciais, P., and Prunet, P.: A new stepwise carbon cycle data assimilation system using multiple data streams to constrain the simulated land surface carbon cycle, Geoscientific Model Development, 9, 3321-3346, 2016, 10.5194/gmd-9-3321-2016.

---

## Author Response (AR1)

We appreciate the comments from two anonymous reviews. We answer below in blue text to each comment listed in black adding the explicit change made in the revised manuscript.

**Reviewer #1**

This paper develops an analysis of the global land C fluxes for three decades. The aim of the paper is to analyse the performance of a data assimilation system, over a historical period. The paper misses clear science questions that would broaden the interest of the study. I lack a clear understanding of the novelty of this work. The conclusions focus on the result that a single decade of data leads to similar results to 3 decades of assimilation, but I am not clear how robust this result is, and how significant it is. I suggest the authors focus more clearly on the novelty of their experimental results. It would be helpful to focus on science questions that are more broadly relevant to the C cycle science community.

We appreciate the reviewer's concerns and are grateful for the comments, which have helped us to formulate the manuscript more clearly. We present three decades of land C flux reanalysis with the aim to better understand the ability of MPI-CCDAS to make decadal projections of the land C cycle. Our approach was to assimilate different time-periods of data to understand the effect of data selection, but also the effect of withholding more recent information. In this way, we can assess the projection of carbon fluxes for the more recent period, and not only against the withheld observations, but also against a model run that has been constrained by this data, in order to disentangle model limitations (i.e. in reproducing an observation that the model was informed about through data assimilation) from prognostic uncertainty (i.e. in failing to reproduce an observation that it can reproduce when given this information). To our knowledge, this is a novel design and focus in data assimilation studies. Previous findings on the prognostic capacity of a CCDAS only looked at short timescales after the assimilation (Scholze et al. 2007; Schürmann et al. 2016), or in the case of Rayner et al. 2011, the authors used a much simpler model ignoring the interacting effects of water, energy and phenology on the carbon cycle predictions. In the revised manuscript, we are more explicit and clearer in the scientific contribution of our results that are relevant to the carbon cycle scientific community.

Our results demonstrate that it is not necessary to ingest more than one decade of observations to improve important features of the global carbon cycle over the following 1-2 decades. In particular, improvements were observed in the long-term trend and seasonal amplitude of atmospheric  $CO_2$  concentrations at station level, as well as in the long-term trend, phenological seasonality and interannual variability of FAPAR. These results provide an insight into the amount of data that is necessary in data assimilation systems to improve the representation of the global carbon cycle components. This information might be decisive for opening the possibility of including newly measured data of other global indicators, such as SIF, that currently only exist for periods of less than a decade.

The new paragraph in the introduction of the revised ms where the aim of the ms is clarified reads:

"The overarching aim of this work is to understand the ability of the MPI-CCDAS v1 to make decadal projections of the land C cycle when the assimilation is confronted to different temporal windows from two observational constraints: FAPAR from remote sensing data and atmospheric CO2 concentrations from the global flask measurements network. For this, we present three decades of modeled land carbon fluxes with the MPI-CCDAS and investigate the effect of withholding information from recent decades in the projected carbon fluxes and the ability of the model to reproduce the

observations during the period of data assimilation. We also analyze trends and seasonal variations in the simulated signals during the periods of the assimilation and compare to independent results to evaluate the model performance. With these results, we gain insights in the number of observations (in terms of decadal scale) necessary in data assimilation systems to improve the representation of the global terrestrial carbon cycle components. These results open the possibility of including newly measured data in DAS that are only available for periods of less than a decade."

It might be helpful to add experiments assimilating just FAPAR or just CO2, and test the outputs against the other (withheld) observational dataset. This experiment might indicate whether the model effectively couples canopy processes with atmospheric concentrations.

We appreciate the suggestion of the reviewer regarding the experiment of only assimilating one of the observational data sets at a time. However, we did not want to repeat this experiment because this has been previously done with the same system in Schürmann et al. 2016. In that work, 5 years of observational data (2005 - 2009) were assimilated in three independent experiments: only FAPAR, only atmospheric CO2 concentrations and the two data sets simultaneously. The results showed that with this time period it was sufficient to demonstrate, in a nutshell, that when assimilating only FAPAR, the average growing season and vegetation seasonality (indicated by FAPAR) was considerably improved in the northern boreal areas. When atmospheric CO2 concentrations were the only assimilated observations, the global gross and net carbon fluxes were overall improved, but the GPP in the tropics was significantly reduced when compared to the GPP after the FAPAR-only assimilation.

While we agree with the reviewer that in principle it would have been interesting to repeat these experiments to analyze their effect on long-term trends, this would require too much computational time to be included in this study and it was not the central focus of this work.

The abstract is too long, it needs to be reduced to 300 words and focused on the key outcome.

We significantly shortened the abstract and remain under 300 words, including only the key outcome regarding using different periods in the observational data during the assimilation.

Writing style is clunky, missing words – for instance the opening sentence is poorly comprehensible: "The observed contemporary in atmospheric CO2 is driven by anthropogenic emissions from fossil fuels and land-use change". I suggest the authors work together to improve the English, reduce sentence lengths, and shorten the more bloated paragraphs.

We realized that we missed the word "increase" in the opening sentence and appreciate the reviewer for pointing this out. The sentence should read: "The observed contemporary **increase** in atmospheric  $CO_2$  ...", we apologize for this. Besides this correction, the revised manuscript has undergone a thorough revision for the English language with the aim of shortening sentences and paragraphs.

**Methods: The global grid is very coarse (8x10) – what are the implications?**

A previous work with the dynamic global vegetation model LPG-DGVM (Müller and Lucht, 2007), demonstrated that the effect of increasing the spatial resolution did not have a strong effect on the simulated regional and global carbon fluxes, whereas temporal dynamics are unaffected. To our knowledge, the effect on the simulated C fluxes after assimilation due to changes in the horizontal spatial resolution of the

model grid cells solely, has not yet been assessed with current data assimilation systems. One reason for this is that the computational cost of such approach is prohibitive. A study using ORCHIDAS (Peylin et al., 2016), suggested that the level of complexity of the ecosystem model in a data assimilation system, including its spatial resolution, does not guarantee improvements in the optimization of C fluxes. Such a conclusion is likely valid for the assimilation of atmospheric observations, but does decrease the ability of the CCDAS to adequately constrain phenology parameters given the mixed phenology in these large pixels. We would like to point out, however, that increasing the spatial resolution to a degree that would allow for a clean identification of PFT-wise parameters, exceeds the resolution of many state-of-the-art forward terrestrial biosphere models. The following paragraph in the revised ms is added in section 2.1 of Methods:

"This horizontal resolution allows computational feasibility and a realistic computational cost for the set of experiments presented in this work. Furthermore, previous evidence has shown that a higher spatial resolution in global vegetation models does not exert a considerable influence in the simulated carbon fluxes at global or regional scales when compared to results obtained with a coarse grid (Müller and Lucht, 2007). The lack of influence to improve the simulated global C fluxes due to changes in the model spatial resolution might also apply to CCDAS (Peylin et al., 2016)."

Why mix such a large spatial grid with such fine temporal scales? Is this valid or required? Why not have a grid that matches TM3?

We are unsure on the meaning of the question by the reviewer, because it combines both spatial and temporal scales. In the CCDAS, we integrate the simulated daily net  $CO_2$  fluxes to monthly scale and transport them with the Jacobian representation of TM3. This approach allows us to account for the non-linear impact of weather anomalies on the surface fluxes, but removes the impact of synoptic atmospheric transport variability on the simulated seasonal and long-term dynamics of atmospheric  $CO_2$  at the monitoring stations. We improved this explanation in the revised manuscript. Although it would be desirable to use MPI-CCDAS on the same grid as TM3, the land-surface model is currently numerically too expensive to allow these higher resolutions.

**Explain what is meant by 'each iteration' (L. 139).**

The term "each iteration" refers to every cycle when the model re-calculates the cost function for the difference between the model parameters and the observational constraint. In section 2.1 of methods we completed the following paragraph for clarification: "During the optimization procedure, a new model trajectory is determined in each iteration (i.e. in every cycle when the model re-calculates the cost function for the difference between the model parameters and the observational constraint), such that energy and mass are conserved through the entire assimilation window (Kaminski and Mathieu, 2017)."

It is unclear how FAPAR data are generated from equation 2, which seems to generate NDVI estimates. We require details on the FAPAR observation operator -I could not find any.

We apologize for the error in Equation 2. In section 2.2 of the revised ms, this is corrected and now it reads:

"Therefore, we used as FAPAR proxy the Global Inventory Monitoring and Modeling System (GIMMS) NDVI product for the period 1982 to 2006 (Tucker et al., 2005), and merged it with the TIP-FAPAR product to provide a longer record of vegetation

greenness. The maximum and minimum NDVI values were rescaled at the pixel level to coincide with those from the TIP-FAPAR for the overlapping periods (i.e., 2003 to 2006) following:

$$FAPAR_{mod} = \frac{NDVI - NDVI_{min,x}}{NDVI_{max,x} - NDVI_{min,x}} \times (TIP_{max,x} - TIP_{min,x}) + TIP_{min,x}$$
(2)

Where x is the period 2003 to 2006 for each data set, NDVI is the full NDVI product from 1982 to 2006, with minimum values given by NDVImin and maximum by NDVImax. TIPmin and TIPmax are the corresponding minimum and maximum values from the TIP-FAPAR product. With this approach, the resulting merged product maintains the maximum and minimum values from TIP-FAPAR while preserving the temporal dynamics of NDVI. The median uncertainty of the available TIP-FAPAR data was considered as the uncertainty for the entire time-series."

**Why are there no FAPAR data after 2006? (1. 278)**

Unfortunately, this is an error during the assimilation procedure. Due to a technical fault, the CCDAS did not consider the remaining four years of data from the original TIP-FAPAR time-series as planned. We discovered this issue only during the post-processing phase, and we wrote it in the final sentence of the paragraph above. However, we believe that our results are still valid, because the main information gain of the CCDAS in terms of phenology stems from the seasonal cycle, with little effect on the overall trends between the three assimilation experiments with different time periods. This comment is added in section 2.1 of methods in the revised manuscript: "Due to a technical failure in the CCDAS, the final FAPARmod product spans only from 1982 to 2006 and the last four years from the TIP-FAPAR product were not included."

And in the Discussion: "The technical error during the assimilation procedure to not include the period from 2007-2010 in the  $FAPAR_{mod}$  product does not influence however the decadal results observed here, because the main information gain of the CCDAS in terms of phenology stems from the seasonal cycle, with little effect on the overall trends between the three assimilation experiments with different time periods."

What are the implications of not including fire emissions? Why were fire products such as GFED not used to provide this input?

Omitting fire fluxes may impair the ability of the MPI-CCDAS to correctly infer the atmospheric growth rate of  $CO_2$  in years with strong contribution of a fire flux, such as the 1998 El Niño. It was not possible to use GFED-like products in this particular assimilation experiment, because for example GFED4 data do not exist prior to 1997, whereas our inversion started in 1982. Adding them only starting in 1997 would have biased the assimilation procedure. In the discussion part of the revised manuscript, and based on what was already presented in the discussion ms, we completed this information to read: "Omitting fluxes in the current model configuration due to fire events may impair the ability of the model to infer the atmospheric growth rate of CO2 associated with El Niño events (Frölicher et al., 2011; Frölicher et al., 2013). One way to overcome the IAV mismatch would be to include fire fluxes in the model by prescribing them from, e.g., the Global Fire Emissions Database (GFED, van der Werf et al., 2010), however the latest version of this data set (Version 4.0) is only available for years from 1997 which is a limiting factor for the timeframe of the simulations in this work. However, the contribution of these interannual variations to the overall CO2 cost function is low in comparison to the signal contained in the

seasonal cycle and deviations in the long-term trend, such that the MPI-CCDAS may simply not be sensitive enough to these aggregate system properties like the response of the tropical carbon cycle to El Niño events given the uncertainty in the atmospheric transport and the observational representation error."

I would like to know more about the process to determine which parameters were selected for optimisation? These seems to mostly phenological variables which will link to FAPAR. I would expect to see other parameters related to C turnover, e.g. mortality rates, decomposition rates.

The choices of parameters were done by an extensive parameter sensitivity study with a large set of MPI-CCDAS model parameters for a wide range of biomes. The retained parameters had a strong effect of the simulated carbon and water fluxes as well as in phenology. This selection process was extensively described in Schürmann et al. 2016. In Table 1 of the discussion and revised ms, we listed the parameters selected for the optimization. While the majority are indeed linked to phenology, we considered also parameters linked to photosynthesis and global parameters that control the land carbon turnover. Those are the last four parameters listed in Table 1: for the heterotrophic respiration the temperature sensitivity to respiration ( $Q_{10}$ ) and a multiplier for initial slow pool ( $f_{slow}$ ) to account for non-steady state conditions at the beginning of the assimilation; for the autotrophic respiration, the leaf fraction of maintenance respiration, and finally an initial atmospheric carbon concentration.

**Results:**

What are the substantial changes in tropical LAI (1. 306) – how do these match in situ measurements?

As shown in Table 1 of the revised ms, the maximum LAI value is one of the optimization parameters and it was prescribed for each PFT. In the case of the tropical evergreen and deciduous trees, this equals to  $7 \text{ m}^2 \text{ m}^{-2}$ . To provide a numerical context to the LAI changes, we summarize in the table below (added in the revised ms as Table A1 in appendix) the mean and maximum LAI values per regions of Fig. 1 for each experiment.

| Region | PRIOR          | ALL            | DEC1           | DEC2           |
|--------|----------------|----------------|----------------|----------------|
|        | (mean / max)   | (mean / max)   | (mean / max)   | (mean / max)   |
|        | $(m^2 m^{-2})$ | $(m^2 m^{-2})$ | $(m^2 m^{-2})$ | $(m^2 m^{-2})$ |
| BE     | 0.61 / 2.29    | 0.60 / 1.94    | 0.70 / 2.42    | 0.69 / 2.42    |
| BW     | 0.31 / 1.62    | 0.30 / 1.44    | 0.35 / 2.01    | 0.35 / 2.02    |
| TNE    | 1.28 / 4.28    | 1.17 / 3.33    | 1.31 / 3.49    | 1.32 / 3.79    |
| TNW    | 1.26 / 3.11    | 1.15 / 2.84    | 1.30 / 3.23    | 1.30 / 3.21    |
| TE     | 1.62 / 3.27    | 1.30 / 2.43    | 1.63 / 3.20    | 1.67 / 3.33    |
| TW     | 2.21 / 3.17    | 1.68 / 2.27    | 2.00 / 2.89    | 2.08 / 3.00    |
| TSE    | 1.54 / 2.72    | 1.43 / 2.51    | 1.86 / 2.77    | 1.83 / 2.68    |
| TSW    | 2.42 / 3.69    | 2.04 / 2.71    | 2.38 / 3.47    | 2.43 / 3.66    |

We also show in the figure below, the average maximum LAI for each experiment for the period 1980-2010.

When we compare the LAI mean values between the experiments and the PRIOR results (Fig. 3 in the discussion manuscript), we observe that the largest change in LAI values was in the tropical west area (TW) comprising Brazil, with a decrease in LAI values of up to 24 % in the ALL experiment with respect to the PRIOR, as a response of the maximum LAI decay in the tropical evergreen PFT (visible in the figure above). We argue that this decrease is a response of a global compensating effect to heterotrophic respiration, leading to the lower GPP tropical value.

Ground based observations in the tropical Amazon-Savanna transition forest have been reported with an annual mean LAI value for the total canopy between 2005 and 2008 of  $7.4\pm0.6 \text{ m}^2 \text{ m}^{-2}$ , and for the seasonal flooded forest a value of  $3.4\pm0.1 \text{ m}^2 \text{ m}^{-2}$ . For the remote sensing data from MODIS, the reported values are  $6.2\pm0.2 \text{ m}^2 \text{ m}^{-2}$  and  $5.8\pm0.3 \text{ m}^2 \text{ m}^{-2}$ , respectively (Biudes et al., 2014).

In the eastern Amazon forest, the remote sensing-based LAI has been reported as 6.2  $m^2 m^{-2}$  from LiDAR, and 4.8  $m^2 m^{-2}$  with a low end of 2.0  $m^2 m^{-2}$  from MODIS (Qu et al., 2011). The maximum LAI values from our model results before and after the assimilation (see table above) fall within the values from MODIS and LiDAR. However, this comparison is robust because of the spatial resolution of the different methods: a coarse model grid cell resolution vs. the resolution of ground-based measurements and the resolution of the remote sensing pixels (50x50 m for ground-based and LiDAR data, and 463 x 463 m for MODIS). This discussion is added

The following paragraph is now added with this information in the discussion of the revised ms:

"Bearing in mind the different spatial resolution of methods (i.e., model grids and remote sensing pixels), a robust comparison between the mean and maximum LAI values before and after the assimilation per region are presented in Table A1 of the Appendix. The results fall within LAI values from MODIS and LiDAR reported in the literature. Ground-based observations in the tropical Amazon-Savanna transition forest between 2005 and 2008 show an annual mean LAI value for the total canopy of 7.4±0.6 m2 m-2 and for the seasonally flooded forest the value of  $3.4\pm0.1 \text{ m}^2 \text{ m}^{-2}$ . For the remote sensing data from MODIS, the reported values are  $6.2\pm0.2 \text{ m}^2 \text{ m}^{-2}$  and  $5.8\pm0.3 \text{ m}^2 \text{ m}^{-2}$ , respectively (Biudes et al., 2014). In the eastern Amazon forest, the remote sensing-based LAI has been reported as  $6.2 \text{ m}^2 \text{ m}^{-2}$  from LiDAR, and  $4.8 \text{ m}^2 \text{ m}^{-2}$  with a low end of 2.0 m2 m-2 from MODIS (Qu et al., 2011)."

In addition, former Figure 3 was removed and replaced into the supplement by a figure (now Fig. S2 presented below) showing the differences between experiments for the average maximum LAI. This is for the sake of keeping the revised manuscript more focused on the main aim, and because the reference to the LAI results were done only sporadically through the manuscript.